# Elucidating the Design Space of Multimodal Protein Language Models

Cheng-Yen Hsieh [* ‡ ♡]   Xinyou Wang [* ◇ ♡]   Daiheng Zhang [† □ ♡]   Dongyu Xue [♡]   Fei Ye [♡]   Shujian Huang [◇]
Zaixiang Zheng [‡ ♡]   Quanquan Gu [♡]

## Abstract

Multimodal protein language models (PLMs) integrate sequence and token-based structural information, serving as a powerful foundation for protein modeling, generation, and design. However, the reliance on tokenizing 3D structures into discrete tokens causes substantial loss of fidelity about fine-grained structural details and correlations. In this paper, we systematically elucidate the design space of multimodal PLMs to overcome their limitations. We identify tokenization loss and inaccurate structure token predictions by the PLMs as major bottlenecks. To address these, our proposed design space covers improved generative modeling, structure-aware architectures and representation learning, and data exploration. Our advancements approach finer-grained supervision, demonstrating that token-based multimodal PLMs can achieve robust structural modeling. The effective design methods dramatically improve the structure generation diversity, and notably, folding abilities of our 650M model by reducing the RMSD from 5.52 to 2.36 on PDB testset, even outperforming 3B baselines and on par with the specialized folding models. Project page and code: `bytedance.github.io/dplm/dplm-2.1`.

## 1  Introduction

Proteins are the molecular machinery of life, encoded by amino acid sequences that fold into intricate three-dimensional structures to perform their biological functions. Existing approaches often treat sequence and structure as separate modalities, relying on disjoint models (e.g., ESM for sequences and AlphaFold for structures (Lin et al., 2023;

Jumper et al., 2021)) that fail to capture the interplay between them. This limitation hinders the ability to jointly model, understand, and generate proteins in a unified framework, which is essential for tasks like protein design, folding, and functional annotation (Lee et al., 2022b; Senior et al., 2020; Lees et al., 2011).

Recent efforts in multimodal protein language models such as ESM3 (Hayes et al., 2024) and DPLM-2 (Wang et al., 2024a) have demonstrated the potential of integrating sequence and structure within a single language model as a unified generative framework. In particular, DPLM-2 is a mulitmodal extension of diffusion protein language model (DPLM, Wang et al., 2024b) with discrete diffusion framework (Austin et al., 2021), which naturally aligns with the discrete nature of protein sequences, enabling it to benefit from large-scale pre-training on sequence databases—a crucial factor for accurate structure prediction (Lin et al., 2022). Beyond sequence modeling, DPLM-2 extends its capabilities by tokenizing 3D coordinates into discrete tokens, thereby enabling direct language modeling of both modalities, hence comprehension and generation of them.

Despite the success, the structure tokenization process introduces structural information loss—obscuring fine-grained geometric relationships critical for accurate protein modeling. Consequently, even such state-of-the-art multimodal PLMs struggle to generate biologically plausible structures for complex tasks like structure folding (Lin et al., 2022) or motif scaffolding (Watson et al., 2023; Yim et al., 2024), where precise structural correlations are crucial. The loss of nuanced variations due to tokenization also degrades the structure diversity in unconditional generation.

In this paper, we systematically explore the key pitfalls and the design space of token-based multimodal protein language models to bridge their limitations on structural modeling. In addition to the structural information loss from tokenization, we identify the primary challenges as the inaccuracies in language model's capability in structure (tokens) prediction, which could not be simply resolved by improving the reconstruction accuracy. We find that index-based structure tokens as supervised labels ignore correlations between semantically similar structure tokens, making the learning process particularly challenging.

In response, we build upon DPLM-2 to advance the design

[*]Equal contribution [†]Core Contributor [‡]Project Lead  [◇]School of Computer Science, Nanjing University [□]Dept. of ECE, Rutgers University [♡]ByteDance Seed (this work was done during Xinyou Wang and Daiheng Zhang's internship at ByteDance Seed). Correspondence to: Quanquan Gu <quanquan.gu@bytedance.com>.

*Proceedings of the 42nd International Conference on Machine Learning*, Vancouver, Canada. PMLR 267, 2025. Copyright 2025 by the author(s).

space spanning improved generative modeling, structure-aware architectures, representation learning, and data exploration (ref. Table 12). We achieve a finer-grained supervision through bitwise discrete modeling and a hybrid approach for data-space modeling. While these methods effectively guide the design of supervision targets, language model-based architectures still lack geometric inductive biases and structural learning objective. To mitigate this, we introduce geometry-aware modules and representation alignment techniques to refine the modeling of higher-order relationship between residues, which is essential as evidenced in protein folding (Jumper et al., 2021). As existing multimodal PLMs are often trained solely on single-chain proteins, we explore the effects of multi-chain proteins (multimer), which introduces richer structural interactions crucial for robust modeling.

We summarize our main contributions as follows:

- We conduct a comprehensive study revealing key pitfalls in structure token-based multimodal protein language models, and systematically elucidate their design space for robust structural modeling.
- Utilizing improved approaches such as bit-wise discrete modeling offers finer-grained supervision, significantly improving structure generative capability.
- Introducing representation-level learning and architectural innovations infuses geometric inductive biases and effectively refines generation diversity.
- We find that multimer and monomer modeling are deeply interconnected and leveraging multimer data advances the structural modeling for both single and multi-chain proteins.
- Our design methods allow multimodal PLMs to achieve robust structural understanding, improving the folding RMSD from 5.52 to 2.36 on the PDB date dataset, outperforming 3B folding baselines with only 650M parameters.

## 2 Revisiting Multimodal Protein Language Models: Capabilities & Constraints

The aim of generative protein modeling is to estimate the underlying distribution $\text{prot} \sim q(\text{prot})$ of all associated moralities of the protein data by learning a probabilistic model $p_\theta(\text{prot})$. Here $\text{prot} = (r_1, r_2, \ldots, r_L)$ denotes a protein with $L$ residues, where each residue $r_i = (s_i, x_i)$ is represented by two major modalities, *i.e.*, $s_i \in \{0, 1\}^{|\mathcal{S}|}$ is a categorical variable for its amino acid type in $\mathcal{S} = \{1, \ldots, 20\}$, and $x_i \in \mathbb{R}^{N_{\text{atoms}} \times 3}$ is the real-value Cartesian coordinates of its residue atoms (we only consider backbone atoms herein, *i.e.*, $[N, C_\alpha, C, O]$ with $N_{\text{atoms}} = 4$). Namely, $p_\theta(\mathbf{s}, \mathbf{x}) = p_\theta(s_1, s_2, \ldots, s_L, \ x_1, x_2, \ldots, x_L)$.

Mutlimodal generative approaches that jointly models structure and sequence can be mainly categorized into two paradigms, *i.e.*, structure-centered diffusion/flow-based models (Campbell et al., 2024b) or sequence-centered language models. The latter is our main focus in this paper, and we will elaborate on this as follows.

### 2.1 Multimodal generative learning for proteins with language models and structure tokenization

Language models (LMs), parameterized by large-scale Transformers (Vaswani et al., 2017) have become the *de facto* choice dominating different domains with scalable and performing expressiveness (OpenAI, 2023). Among them, protein LMs have been serving as one of the AI foundation for protein sequence learning (Rives et al., 2019; Lin et al., 2022) and generation (Nijkamp et al., 2022; Hayes et al., 2024).

**DPLM.** Diffusion protein language model (DPLM, Wang et al., 2024b), in particular, shows excelling performance in both generation and representation learning of protein sequences, and even structures thanks to its recent multimodal extension DPLM-2 (Wang et al., 2024a). The family of DPLMs grounded in *absorbing* discrete diffusion framework (Austin et al., 2021; Zheng et al., 2023a), which is characterized by a forward and backward Markov process. Let $\text{Cat}(\mathbf{x}; \mathbf{p})$ be a categorical distribution on protein sequence $\mathbf{y}$ parameterized by a vector $\mathbf{p}$ on $(|\mathcal{V}| - 1)$-dimensional probability simplex. The forward process of discrete diffusion defines a Markov process governed by the transition kernel $q(\mathbf{x}^{(t)}|\mathbf{x}^{(t-1)}) = \text{Cat}(\mathbf{x}^{(t)}; \beta_t \mathbf{x}^{(t-1)} + (1 - \beta_t)\mathbf{q}_{\text{noise}})$ that gradually perturb the data $\mathbf{x} \sim q(\mathbf{x})$ into a stationary distribution $\mathbf{x}^{(T)} \sim \mathbf{q}_{\text{noise}}$. The learned *backward* process $p_\theta(\mathbf{x}^{(t-1)}|\mathbf{x}^{(t)})$ reversely denoises the $\mathbf{x}^{(T)}$ towards the data distribution $\mathbf{x}$, which is typically optimized by the variational bound of the log-likelihood (Ho et al., 2020). The learning objective of absorbing diffusion can be simplified into weighted cross-entropiess, resembling masked language modeling at arbitrary noise levels:

$$
\begin{aligned}
\mathcal{J}_t &= \mathbb{E}_{q(\mathbf{x})} - \text{KL}\big[q(\mathbf{x}^{(t-1)}|\mathbf{x}^{(t)}, \mathbf{x})\|p_\theta(\mathbf{x}^{(t-1)}|\mathbf{x}^{(t)})\big] \\
&= \mathbb{E}_{q(\mathbf{x})}\big[\lambda^{(t)}\textstyle\sum_{1 \leq i \leq L} b_i(t) \cdot \log p_\theta(x_i|\mathbf{x}^{(t)})\big],
\end{aligned}
$$

where $\lambda^{(t)}$ is a weighting coefficient induced from the specific noising schedule. For inference, DPLM is able to generate amino acid sequences by the reverse iterative denoising process in a *mask-predict* manner, which starts from an all-mask sequence and iterates towards a synthesized sequence. At time $t$, it first generates $\tilde{\mathbf{x}}^{(0)}$ from $p_\theta(\cdot|\mathbf{x}^{(t)})$, then a less noisy $\mathbf{x}^{(t-1)}$ is sampled by $q(\cdot|\mathbf{x}^{(t)}, \mathbf{x}^{(0)} = \tilde{\mathbf{x}}^{(0)})$.

**DPLM-2: A multimodal extension of DPLM.** To facilitate structure learning in language models, DPLM-2 (Wang et al., 2024a) extends DPLM by introducing a token-based latent representation for protein structure. This is achieved via a two-stage approach: (1) a structure tokenizer firstly learns to convert $\mathbf{x} \in \mathbb{R}^{L \times N_{\text{backb}} \times 3}$, the 3D coordinates of the protein backbone into a discrete structure token sequence, denoted as $\mathbf{z} = (z_1, z_2, \ldots, z_L) \in \{0 \ldots |\mathcal{Z}|\}^L$, where each token $z_i$ represents a local structural element of the $i$-th

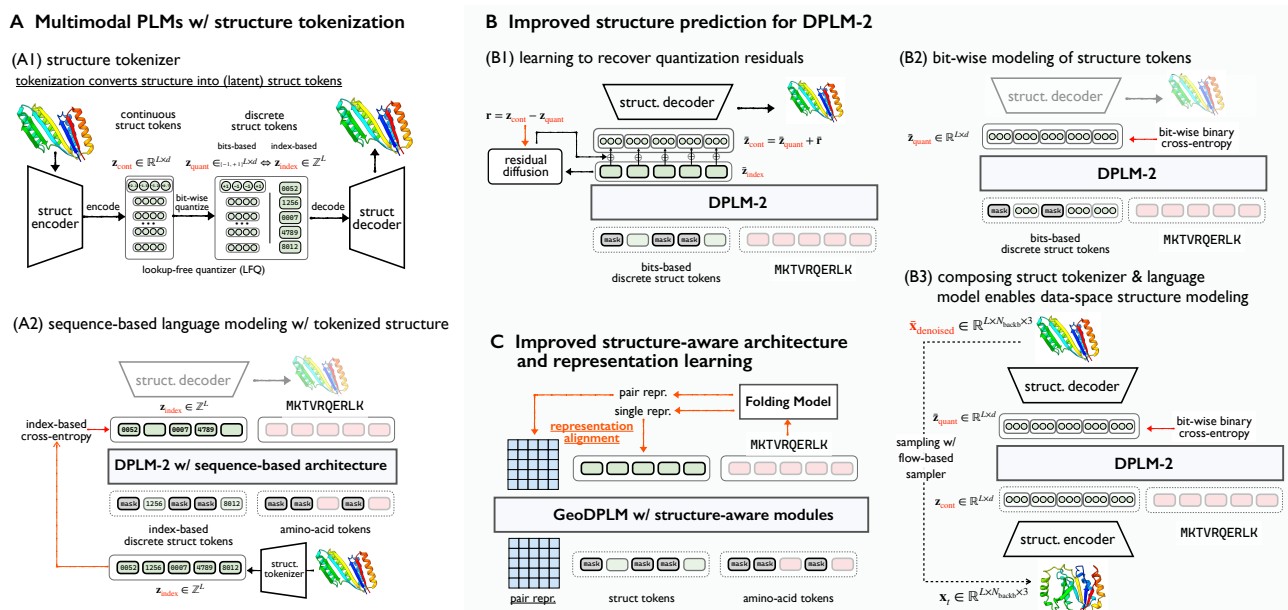

*Figure 1.* **Illustration of multimodal protein language models with structure tokenization, and our proposed improved approaches.** (A): Structure tokenizer and multimodal protein language modeling within sequence-based architecture. We use DPLM-2 (Wang et al., 2024a) as a case study herein; (B) Improved structure prediction approaches for DPLM-2, including quantization residual learning, bit-wise discrete modeling and combining structure encoder, decoder and language model as a structure denoising model for data-space structure modeling. (C) Improved structure-aware model architecture and representation learning aligning to folding models.

residue and $|\mathcal{Z}|$ is the codebook size; (2) given tokenized structure, DPLM-2 processes mulitmodal input, and then performs joint language modeling of the structure token sequence $\mathbf{z}$ with the corresponding amino acid sequence $\mathbf{s}$ for the same protein. The training objective of DPLM-2 hence becomes:

$$\mathcal{J}_t = \mathbb{E}_{q(\mathbf{x},\mathbf{z})}\Big[\lambda^{(t)}\textstyle\sum_i b_i(t)\Big(\log p_\theta(s_i|\cdot) + \log p_\theta(z_i,|\cdot)\Big)\Big]$$

**Structure tokenization.** Any vector-quantization based approach can be studied for tokenizing protein atomic structure into structure tokens (Van Den Oord et al., 2017a). Specifically, DPLM-2 employs an LFQ-based structure tokenizer (Yu et al., 2023), which is the state-of-the-art approach for visual tokenization. This LFQ tokenizer can be summarized as follows:

$$\mathbf{x} \xrightarrow{\text{encoder}} \mathbf{z}_{\text{cont}} \xrightarrow[\text{quantize}]{\text{dimension-wise}} \mathbf{z}_{\text{quant}} \Leftrightarrow \mathbf{z}_{\text{index}} \xrightarrow{\text{decoder}} \tilde{\mathbf{x}},$$

where (1) a structure encoder encodes backbone 3D coordinates $\mathbf{x} \in \mathbb{R}^{L \times N_{\text{backb}} \times 3}$ into invariant features as continuous structure tokens $\mathbf{z}_{\text{cont}} \in \mathbb{R}^{L \times D}$, (2) an LFQ module quantizes $\mathbf{z}_{\text{cont}}$ independently dimension-wise into bits-based (binary) discrete structure tokens $\mathbf{z}_{\text{quant}} \in \{-1, +1\}^{L \times D}$, which can be converted to decimal index-based discrete structure tokens $\mathbf{z}_{\text{index}} = \sum_k^D \mathbf{1}(z_{\text{quant}}^{[k]} > 0) \cdot 2^{k-1} \in \{0...|\mathcal{Z}|\}^L$; and (3) a structure decoder reconstructs 3D coordinates from the discrete tokens.

## 2.2 The pitfalls of modeling over tokenized structures

Albeit being the key enabler to multimodal protein language models, using discrete structure tokens to represent structural information also limits the model's ability to capture structural details accurately. This trade-off represents an important challenge in the current field of multimodal protein language models. To understand this, we conduct in-depth study regarding structure tokenization and structure prediction by language models. We highlight our (**O**)bservations along with their implications as follows:

**(O1): Structure tokenization results in information loss.** Vector quantization converts latent features of continuous structure tokens ($\mathbf{z}_{\text{cont}}$) from the structure encoder into discrete structure tokens ($\mathbf{z}_{\text{quant}}$), discarding residual information ($\mathbf{z}_{\text{cont}} - \mathbf{z}_{\text{quant}}$). As shown in Table 1, however, applying quantization significantly amplifies reconstruction errors (RMSD $1.31 \nearrow 1.98$ & TMscore $0.97 \searrow 0.93$). This indicates that quantizing continuous tokens into discrete tokens inevitably results in loss of fidelity hence detailed structural accuracy. This **suggests** that learning to recover the lost residuals—particularly as a refinement step—could enhance structure prediction accuracy.

*Table 1.* **Effects of feature quantization on structure tokenizer reconstruction.**

| Latent feature | Struct token type | Reconstruction | |
|---|---|---|---|
| | | RMSD↓ | TMscore↑ |
| $\mathbf{z}_{\text{cont}}$ | (pre-quantized) continuous token | **1.3127** | **0.9733** |
| $\mathbf{z}_{\text{index}} \Leftrightarrow \mathbf{z}_{\text{quant}}$ | (quantized) discrete token | 1.9806 | 0.9385 |

**(O2): High reconstruction accuracy does not guarantee better structure generative performance in language models, while a significant gap remains in between.** We compare the impact of different protein structure tokenizers on reconstruction and generation tasks. Specifically, we select tokenizers from DPLM-2 and ESM3, training separate DPLM-2 variants with the same architecture but using their respective structure token codebook. These models are evaluated on the CAMEO 2022 test set for both reconstruction and protein folding performance. As shown in Table 2, the ESM3 tokenizer achieves superior reconstruction accuracy (RMSD: 0.72, TMscore: 0.99), outperforming the DPLM-2 tokenizer. However, the model trained with the DPLM-2 tokenizer's codebook exhibits stronger protein folding performance. This suggests that while reconstruction accuracy sets an upper bound on generation quality, the substantial gap between the two highlights the critical role of the language model's generative capability in structure prediction. This **suggests** that, given that mild improvement in reconstruction do not necessarily translate into better generation, greater emphasis should be placed on improving structure-aware generative modeling and architectural design.

*Table 2.* **Tokenizer reconstruction *vs.* language model generation.** Evaluation of folding on CAMEO 2022.

| Tokenizer | Reconstruction | | Generation | |
|---|---|---|---|---|
| | rRMSD↓ | rTMscore↑ | RMSD↓ | TMscore↑ |
| DPLM-2 | 1.9806 | 0.9385 | **7.7025** | **0.7936** |
| ESM3 | **0.7248** | **0.9912** | 8.4424 | 0.7924 |

**(O3): Index-based structure tokens? Multimodal PLM gets them miserably wrong in structure prediction.** Table 3 shows that direct index prediction is highly inaccurate (0.0864 accuracy on CAMEO). However, the structural evaluation metrics (RMSD, TMscore) indicate that the generated structures do not completely collapse, suggesting that despite the coarse-grained supervision, the model still captures some underlying relationships between indices. This learning process, however, remains highly challenging: since each index is derived from multiple quantized bits, even small changes at the bit level can result in drastically different indices. This issue becomes even more problematic as the codebook size increases, further exacerbating the difficulty of direct index prediction. In contrast, when evaluated at the bit-based level, prediction accuracy reaches 0.7720 on CAMEO, which aligns more closely with structural evaluation metrics. This **suggests** that while the model struggles to recover exact indices, it effectively captures structural patterns at the bit level.

**Concluding Remarks.** Given these observations, we identify the primary bottlenecks of token-based multimodal protein language models as tokenization loss (**O1**) and ineffective structure modeling in sequence-based architectures (**O2&O3**). To address these, we introduce improved generative approaches for structure prediction (§3) and enhanced

*Table 3.* **Language model structure token prediction accuracy.** Index-based *vs.* bits-based evaluation on structure folding.

| Model | Testset | Struct Token Acc↑ | | Struct Eval Metric | |
|---|---|---|---|---|---|
| | | index | bit | RMSD↓ | TMscore↑ |
| DPLM-2 index-based | CAMEO 2022 | 0.0864 | 0.7720 | 7.7025 | 0.7936 |
| | PDB date split | 0.1188 | 0.7932 | 5.3071 | 0.8306 |
| DPLM-2 BIT-based | CAMEO 2022 | **0.1258** | **0.7958** | **6.4028** | **0.8380** |
| | PDB date split | **0.2641** | **0.8648** | **3.2213** | **0.9043** |

architectural designs with better geometric awareness and representation learning (§4).

## 3 Improved Structure Prediction

In this section, we present several improvements aimed at enhancing the accuracy and detail of protein structure modeling. These approaches build upon the initial structure tokenization and aim to improve predictions by addressing challenges by introducing methods for recovering tokenization losses, bridging discrete and continuous tokens, and enabling direct data-space modeling.

### 3.1 Recovering Tokenization Loss by Learning Quantization Residuals with RESDIFF

In the structure tokenizer, the vector quantizer module converts encoded structure features into discrete structure token features, fundamentally clustering similar local environments into identical token. However, according to **O1**, this process inherently introduces lossy compression, the residuals—the differences between the original and quantized features—are lost during this process, eliminating fine-grained structural details. The primary principle of the solution is that we need to recover and preserve the high-frequency variation that gets lost during tokenization.

To address this issue, we can utilize continuous generative modeling, such as diffusion or flow-based models, to learn to recover these residuals. Specifically, inspired by Tang et al. (2024), we introduce a light-weight diffusion module, *i.e.*, RESDIFF, to predict the residual information. Formally, let $\mathbf{r} = \mathbf{z}_{\text{cont}} - \mathbf{z}_{\text{quant}}$ represent the quantization residuals, we aim to learn a light-weight generative model (Ho et al., 2020; Nichol & Dhariwal, 2021) to predict the residual $\mathbf{r}$ conditioned on the hidden states of language model $\mathbf{h}$ and discrete structure tokens $\mathbf{z}$. The training loss is to minimize

$$\mathcal{L}_\phi = \mathbb{E}_{q(\mathbf{r}), \epsilon \sim \mathcal{N}(0, \mathbf{I}), t} \left[ ||\epsilon - \epsilon_\phi(\mathbf{r}_t, t, \mathbf{h}, \mathbf{z}_{\text{quant}})||_2^2 \right].$$

As illustrated in Fig. 1(B1), the protein structure generation process now begins by generating discrete structure tokens $\mathbf{z}_{\text{index}}$ that capture the overall topology, and then these tokens and the hidden states of language model $\mathbf{h}$ are fed into to RESDIFF to generate the missing residuals $\mathbf{r}$. These residuals would be added up to the structure token to recover continuous structure tokens, *i.e.*, $\mathbf{z}_{\text{cont}} = \mathbf{z}_{\text{quant}} + \mathbf{r}$, closer to the features produced by the structure encoder. Finally this $\mathbf{z}_{\text{cont}}$ is decoded to atomic structure.

*Table 4.* **Evaluation of improved approaches for structure prediction based upon DPLM-2.** Folding SFT: supervised fine-tuning with folding objective.

| Models | CAMEO 2022 | | PDB date split | |
|---|---|---|---|---|
| | RMSD↓ | TMscore↑ | RMSD↓ | TMscore↑ |
| ESMFold (3B) (Lin et al., 2022) | 3.9900 | 0.8500 | 2.8400 | 0.9300 |
| MultiFlow (Campbell et al., 2024a) | 17.8400 | 0.5000 | 15.6400 | 0.5300 |
| ESM3 (1.4B) (Hayes et al., 2024) | 6.3300 | 0.8400 | 4.9003 | 0.8653 |
| DPLM-2 (650M) | 7.7025 | 0.7936 | 5.3071 | 0.8306 |
| DPLM-2 + RESDIFF | 7.2881 | 0.8087 | 5.1072 | 0.8430 |
| DPLM-2 (BIT-based) | 6.4028 | 0.8380 | 3.2213 | 0.9043 |
| DPLM-2 (BIT-based) + RESDIFF | 6.1781 | 0.8428 | 3.0168 | 0.9076 |
| DPLM-2 (BIT-based) + FM | 6.1825 | 0.8414 | 2.8697 | 0.9099 |
| DPLM-2 (BIT-based) + FM + RESDIFF | 6.0765 | 0.8456 | 2.7884 | 0.9146 |
| *w/* folding SFT | 5.8472 | 0.8442 | 2.3698 | 0.9270 |
| DPLM-2 (3B) *w/* folding SFT | 5.9832 | 0.8443 | 3.1502 | 0.9012 |

**Results.** We evaluate the structure prediction performance on the folding task. As shown in Table 4, the residual diffusion module is capable of improving the structural prediction accuracy by refining fine-grained structural variations based on language model predictions. Moreover, we observe that the residual diffusion module is model-agnostic, showing consistent performance improvements across different DPLM-2 variants. The Fig. 7 demonstrates that the residual diffusion module performs fine-grained refinements on the local structure, optimizing interatomic distances to facilitate the formation of plausible secondary structures.

### 3.2 Bridging Discrete and Continuous Structure Tokens with BIT-based Language Modeling

Discretizing protein structures into index-based tokens enables multimodal PLMs to perform structural modeling but introduces significant challenges, as discussed in **O3**. Since DPLM-2 employs LFQ tokenizer, its bit-level representation provides more informative supervision signals, as reflected by **O3** and Table 3, where bit-level accuracy is more indicative of generation quality. To bridge this gap, we aim to perform language modeling of the bit-based feature of structure tokens instead of their indices, as illustrated in Fig. 1(B2), To this end, inspired by Han et al. (2024), we make the most of DPLM-2's LFQ tokenizer, which already operates at the bit level, which quantizes each dimension independently, preserving more structural details while remaining compatible with PLMs' discrete supervision. It hence becomes $K$ binary classifications to predict each bit of $K$-bit structure token, instead of the original $2^{K-1}$-way classifications. This greatly reduces the learning challenges and thus improves generative accuracy. As such, the training objective with bits-based structure modeling is accordingly modified as:

$$\mathcal{J}_t^{\text{bit}} = \mathbb{E}_{q(\mathbf{x},\mathbf{z})}\left[\lambda^{(t)}\sum_i b_i(t)\left(\log p_\theta(s_i|\cdot) + \sum_k \log p_\theta(z_{i,\text{quant}}^{[k]}|\cdot)\right)\right]$$

**Results.** As shown in Table 3 and Table 4, the bit-level supervised DPLM-2 achieves significant accuracy improvements across both index-level and bit-level, while substantially reduced structural deviation from ground truth (re-flected in improved RMSD and TM-score), particularly on the PDB date test set. This suggests that the fine-grained bit-level supervision signals are more suitable for model to learn, enabling the model to capture structural patterns more effectively, which enhances the latent structural modeling.

### 3.3 A Hybrid Generative Approach Enables Direct Data-space Modeling

Table 1 highlights a key limitation of discrete structure tokenization: while it efficiently captures high-level topology and enables co-generation, the transition to a latent-space language model inevitably sacrifices atomic-level details.

However, this transition inherently disentangles geometric modeling from sequence-based generative modeling—with the structure tokenizer serving as an encoder-decoder in data space and the language model operating in the latent space of structure tokens. While this separation introduces information loss, it also presents an opportunity: the combination of the structure encoder, language model, and decoder as a whole effectively functions as a denoising model, capable of refining structure in atomic coordinates. In such a way, we define a structure denoising model $\mathbf{x}_\theta(\mathbf{x}_t, t) : \mathbf{x}_t \mapsto \bar{\mathbf{x}}_{\text{denoised}} \triangleq \text{decoder} \circ \text{PLM} \circ \text{encoder}(\mathbf{x}_t)$. This insight allows us to seamlessly integrate this structure denoising model into a generative framework. Inspired by AlphaFlow repurposing folding models as flow-based generative models (Jing et al., 2024), we incorporate this newly composed structure denoiser $\mathbf{x}_\theta$ into a flow-based sampler with Euler integrator, where each Euler step interpolates $\mathbf{x}_s \leftarrow \frac{s-t}{1-t} \cdot \mathbf{x}_\theta(\mathbf{x}_t, t) + \frac{1-s}{1-t} \cdot \mathbf{x}_t$ up to a Kabsch alignment of $\bar{\mathbf{x}}_{\text{denoised}}$ against $\mathbf{x}_t$, treating it as a denoising process on data-space structure generation. We finetune such a model with flow matching (FM). This hybrid approach enables direct sampling in data space while preserving the scalability of discrete tokenization, ultimately improving atomic-level accuracy in protein modeling.

**Results.** Table 4 demonstrates that direct data-space sampling with flow matching is capable of enhancing the structure generation on the folding task, while supervision with the folding objective can bring further improvement. We observe that the folding performance is on par with or even outperforms the strong baseline ESMFold, particularly on the PDB date split. This demonstrates that the hybrid structure modeling method can take the both worlds of the scalability of discrete tokenization enhanced by language model and the accurate atomic-level geometric modeling, resulting in a superior protein structure generation performance.

## 4 Improved Structure-aware Architecture and Representation Learning

As evidenced in protein folding (Jumper et al., 2021; Lin et al., 2023), the intricate nature of protein structures demands methods that capture higher-order relationship between residues beyond simple sequence-based models.

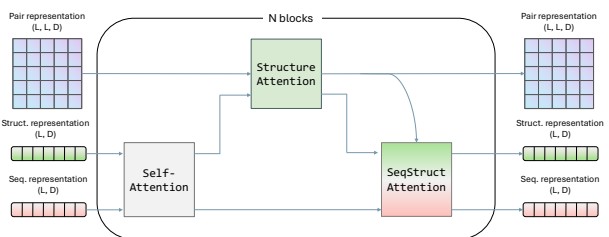

Figure 2. **Integrate geometry-aware modules into multimodal protein language models (PLMs).** We integrate geometric modules into encoder blocks of PLMs, including structure attention (top) and a SeqStruct attention (bottom right). We provide architecture details in Figures 3 and 4 and the appendix.

While bit-based modeling offers effective guide the design of supervision targets, sequence-based models still (1) lack the geometric inductive biases and (2) structural learning objective that might be needed to capture the complexity of residual interactions. To address these limitations, we introduce *geometric modules* and *representation alignment* (*REPA*) (Yu et al., 2024) to enhance the understanding of residue-residue interactions and structural diversity.

### 4.1 GeoDPLM: Geometry-aware Model Architecture

Inspired by PairFormer in AlphaFold3 (Abramson et al., 2024), we introduce GeoDPLM with a newly added *geometric modules* that operates on compact 2D pair representations to capture pairwise spatial dependencies of residues. As shown in Figure 2, we use a structure attention to independently refine structure (single) representations and pair representations through transition and triangle operations, followed by the Seqstruct attention to blend pair representations with both sequence and structural (single) representation.

**Component-wise analysis of GeoDPLM on structure folding.** Table 5 provides a component-wise study on each geometry-based module, which could be referred in Figures 3 and 4. Introducing 2D pair representation maps into DPLM-2 effectively improves structure prediction, reducing the folding RMSD from 7.703 to 7.244 RMSD and increases the TMscore to 0.8339. When models are trained without the folding SFT objective $\max_\theta \log p_\theta(\mathbf{x}|\mathbf{s})$, the inclusion of transition layers for structure representations proves critical for achieving better structure predictions. SeqStruct attention, which combines pair representations with sequence information, achieves minimal improvement in metrics, indicating that solely incorporating pair information into structure representations is sufficient. In contrast to the common practice in structure folding, we find that triangle update and attention operations do not yeild notable benefits. For simplicity, in subsequent experiments, we refer to the model with logits bias and pair representation transition layers as GeoDPLM (Base).

**Training efficiency.** Figure 5 presents the correlation between training speed and structure prediction performance

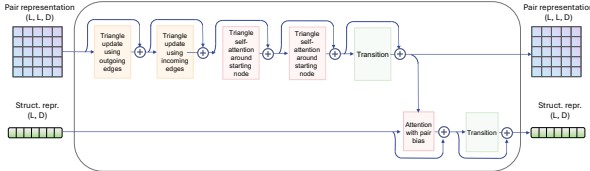

Figure 3. **Structure attention module.** We introduce geometric-aware operations such as triangle operations and logit bias (Jumper et al., 2021). L denotes the number of residues and D is the feature dimension, where we select 128 for pair representation and 1280 for structure representation.

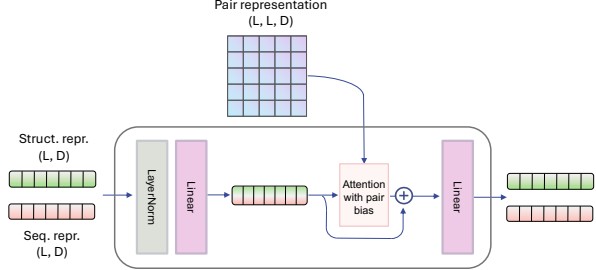

Figure 4. **Seqstruct attention module.** We concatenate structure and sequence representations along the feature dimension. Residual connections between inputs and outputs are omitted here.

on CAMEO2022 for each ablation variation in Table 5. We report the average training speed every 10k training steps, and use marker colors to reflect diversity. Among all operations, triangle operations are the most computationally intensive, with triangle update (TU) and attention (TA) dramatically reducing the training efficiency by 3.9x and 6.8x, respectively. The attention mechanism involving pair representations as logits bias in GeoDPLM (Base) causes a slight decrease on training speed compared to DPLM-2. Using transition layers for structure representations, which significantly boosts both structure modeling and generation diversity, has minimal impacts on training speed.

### 4.2 Representation Alignments to Folding Model

*REPA* benefits structure generation potentially by (1) overcoming the limitations of discrete tokens and (2) addressing the key challenge in training diffusion models. Unlike discrete supervision that enforces sharp targets, *REPA* enables smooth, informative and high-dimensional learning, preserving finer structural nuances. Meanwhile, Yu et al. (2024) has identified the primary challenge of training diffusion models as learning high-quality representations. To address this, we adopt *REPA* by aligning the representations of the protein language model to transfer meaningful structural semantics from specialized folding model, where we use ESMFold (Lin et al., 2023) in this paper for efficiency of massvie inference while other models such as AlphaFolds are also compatible. Originally successful in vision tasks, we find that REPA effectively improves the structural diversity of generated proteins (Figure 6).

**REPA setup.** We select ESMFold (Lin et al., 2023) as the target representation encoder due to its computational

*Table 5.* **Ablation: geometry-aware modules.** We ablate each component in Figures 3 and 4 by studying their effects on folding. P.Bias & Tran: pair bias and transition layer for pair representation. S. Tran: transition layer for structure representation. Tri. Up. & Attn.: triangle update and attention layers. We name each ablation variation in the parenthesis such as (base). SFT: supervised fine-tuning using folding objective.

| Methods | Structure Attention | | | | Seqstruct Attn. | SFT | PDB date split | | CAMEO 2022 | |
|---|---|---|---|---|---|---|---|---|---|---|
| | P. Bias & Tran. | S. Tran. | Tri. Up. | Tri. Attn. | | | RMSD↓ | TMscore↑ | RMSD↓ | TMscore↑ |
| DPLM-2 | × | × | × | × | × | × | 5.521 | 0.8287 | 7.703 | 0.7936 |
| GeoDPLM (Base) | ✓ | | | | | | 4.823 | 0.8521 | 7.244 | 0.8128 |
| (ST) | ✓ | ✓ | | | | | **3.883** | **0.8857** | **6.550** | **0.8339** |
| (TU) | ✓ | ✓ | ✓ | | | | 4.837 | 0.8598 | 7.197 | 0.8255 |
| (TA) | ✓ | ✓ | | ✓ | | | 4.415 | 0.8690 | 6.973 | 0.8210 |
| (SSA) | ✓ | | | | ✓ | | 4.040 | 0.8841 | 7.158 | 0.829 |
| DPLM-2 | × | × | × | × | × | ✓ | 3.347 | 0.9008 | 6.612 | 0.8233 |
| GeoDPLM (Base) | ✓ | | | | | ✓ | 3.165 | 0.9046 | **6.227** | **0.8414** |
| (ST) | ✓ | ✓ | | | | ✓ | **3.021** | **0.9062** | 6.288 | 0.8393 |
| (TU) | ✓ | ✓ | ✓ | | | ✓ | 3.639 | 0.8903 | 6.877 | 0.8322 |
| (TA) | ✓ | ✓ | | ✓ | | ✓ | 3.863 | 0.8790 | 6.393 | 0.8340 |
| (SSA) | ✓ | | | | ✓ | ✓ | 3.134 | 0.9054 | 6.329 | 0.8379 |

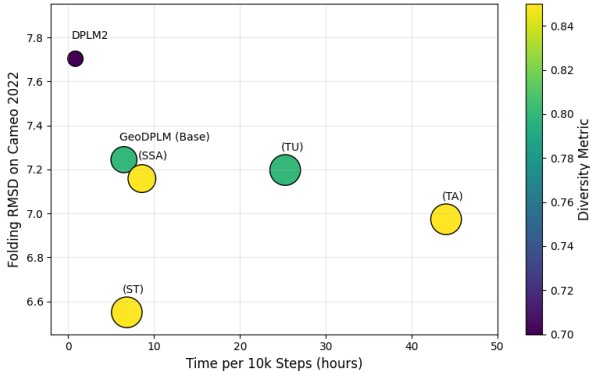

*Figure 5.* **Training efficiency of geometric designs.** We analyze the correlation between training speed and folding RMSD for each ablation variation in Table 5. Color represents generation diversity, while marker size indicates model parameters. Adding transition layers for structure representations (ST) improve diversity and folding without reducing much training speed.

efficiency and open-source availability compared to other folding models (Jumper et al., 2021; Abramson et al., 2024). Using the protein sequences in our training data, we precompute the structure and pair representations from the folding trunk of ESMFold. We run three recycling iterations within the folding trunk to ensure the quality of representations. Instead of aligning embeddings from a certain layer as done in Yu et al. (2024), we apply a learnable `nn.Parameter` followed by a softmax function as the weight to ensemble representations across all layers. We follow Yu et al. (2024) to apply a 3-layer MLP to project the representations before aligning them with the target representations through negative cosine similarity.

**Compatibility with different architectures.** Table 6 presents the effects of REPA on structure prediction. REPA improves structure folding on both PDB date split and Cameo datasets, highlighting its ability to go beyond refining the generation process. Importantly, we observe that

*Table 6.* **Representation alignment improves structure prediction.** REPA is compatible with both language model-based architectures (DPLM-2) and geometric designs (GeoDPLM).

| Methods | PDB date split | | CAMEO 2022 | |
|---|---|---|---|---|
| | RMSD↓ | TMscore↑ | RMSD↓ | TMscore↑ |
| DPLM-2 | 5.521 | 0.8287 | 7.703 | 0.7936 |
| *w REPA* | 4.919 | 0.8508 | 7.344 | 0.8046 |
| GeoDPLM | 4.823 | 0.8521 | 7.244 | 0.8128 |
| *w REPA* | **4.340** | **0.8671** | **7.058** | **0.8217** |

REPA is versatile, being compatible with both language model-based architectures (DPLM-2) and those incorporating geometric designs (GeoDPLM). Applying REPA on DPLM-2 effectively increases both folding RMSD and TM-score, suggesting that additional learning signals can play a critical role in improving architectures that lack geometric inductive biases.

**Better structure-aware model architecture and representation learning boosts structure generation diversity.** Protein language models often suffer low diversity in generated structures (Wang et al., 2024a), potentially due to the lack of inductive biases to capture structural interactions. Figure 6 shows structure generation diversity, where for each protein length, we generate 40 samples and quantify the diversity by the normalized number of sample clusters identified by FoldSeek (Van Kempen et al., 2024). These results show that appropriate geometric architecture designs (GeoDPLM) could effectively improve the generation diversity. Aligning representations to structure folding model further significantly diversifies generated structures, showing its effectiveness in refining generation process.

## 5 On the Orthogonality of Design Methods

Building on the individual analysis of each design method in the preceding sections, we now examine the interactions of these designs by combining them in a unified setting, as

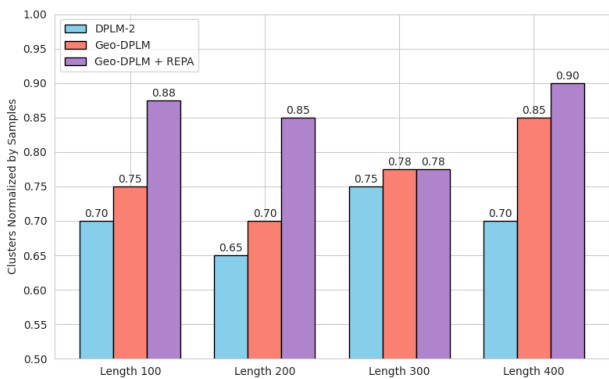

*Figure 6.* **Effects on generation diversity.** Geometric designs and structure representation alignment with the folding model (Lin et al., 2023) significantly improve the low generation diversity of the multimodal PLM. Diversity is normalized by the number of generated samples.

*Table 7.* **Analysis of orthogonality**. We analyze the compatibility of design methods when combined, with the recommended setting highlighted. *All*\* denotes the combination of all methods: Geo, Bit, FM, REPA, ResiDiff, and SFT.

| Models | PDB date split | | CAMEO 2022 | | Uncond. Gen. |
|---|---|---|---|---|---|
| | RMSD↓ | TMscore↑ | RMSD↓ | TMscore↑ | Diversity↑ |
| DPLM-2 (650M) | 5.307 | .8306 | 7.703 | .7936 | 0.700 |
| Bit | 3.221 | .9043 | 6.403 | .8380 | 0.825 |
| Bit + FM | 2.870 | .9099 | 6.183 | .8418 | 0.525 |
| Bit + FM + ResDiff | 2.788 | .9146 | 6.077 | .8456 | 0.525 |
| *w/* SFT | 2.370 | .9270 | 5.847 | .8442 | - |
| Geo + Bit | 2.551 | .9254 | 5.955 | .8520 | 0.900 |
| Geo + Bit + FM | 2.443 | .9261 | 6.172 | .8404 | 0.575 |
| Geo + Bit + REPA | 2.507 | .9264 | 6.192 | .8412 | 0.875 |
| *w/* SFT | 2.404 | .9322 | 5.754 | .8424 | - |
| *All*\* | 2.379 | .9297 | 6.200 | .8398 | - |

shown in Table 7. For models fine-tuned with the folding SFT objective, we skipped the evaluation of unconditional co-generation. We discuss the final recommended setting and the orthogonality of designs below.

**SFT**. Folding SFT improves the structure folding but sacrifices the model's ability for multimodal co-generation, as it is fine-tuned specifically for the folding task.

**The recommended setting: Geo + Bit-based modeling**. Geometric architectures are compatible with bitwise modeling and their combinations achieve comparable results with models finetuned with folding SFT on structure folding, and further obtains an effective improvement on unconditional generation quality & diversity. This setting is also effective in terms of training efficiency as it avoids additional computational overhead from other methods like FM and REPA.

**REPA and Bit-based modeling.** Both REPA and bit-based modeling enhance structure folding and generation diversity. Meanwhile, their combinations do not lead to further improvements. We suggest that this is because REPA and bit-based modeling both help through enabling smooth and high-dimensional learning signals compared to index-based

*Table 8.* **Statistics of PDB-Multimer.** We curate a dataset of multichain proteins from PDB to analyze their effects on structural modeling.

| Dataset | # proteins Train / Val | # chains | Protein Length | Chain Length |
|---|---|---|---|---|
| PDB-Multimer | 11614/291 | 2.88 ± 1.66 | 661.57 ± 416.37 | 229.39 ± 167.00 |

*Table 9.* **Effects of monomer data on tokenizer reconstruction for multimer.** Scaling *monomer* data significantly improves the structure tokenizer reconstruction on PDB-Multimer, suggesting the relevance between multimer and monomer modeling. We also provide results of monomer on CAMEO dataset.

| Training Data | Size | PDB-Multimer | | CAMEO 2022 | |
|---|---|---|---|---|---|
| | | RMSD↓ | TMscore↑ | RMSD↓ | TMscore↑ |
| PDB & SwissProt | 200K | 9.973 | 0.694 | 2.589 | 0.930 |
| AFDB_Rep | +1.2M | **6.873** | **0.784** | **2.245** | **0.938** |

discrete tokens, hence their non-orthogonal effects.

**Hybrid modeling and geometric modules**. FM effectively improves folding, but the benefits diminish with geometric modules, and can reduce generation diversity due to its ODE nature. However, benefitting from the same nature, FM accelerates the sampling process by requiring 10x fewer sampling steps.

**ResDiff**. Similar to the results in the paper, ResDiff does not bring a significant boost to folding metrics. The major benefit of ResDiff is to provide a finer local structure as discussed in the Figure 7.

We include a summary table linking to the motivation and findings of all design methods in Table 12 of the Appendix.

## 6 Structure Data: Multimer Exploration

Despite the advancements in modeling, the scarcity of protein structure data remains a challenge for developing robust multimodal protein language models. In this section, we extend our data coverage to include multimer, proteins that consist of multiple chains, since multimer data presents diverse structural arrangements and interaction scenarios, which are essential for developing a more general multimodal model. Notably, most existing protein language models have been trained solely on single-chain proteins (monomer). In Table 8, we introduce our PDB-Multimer dataset. We examine the relevance and gap between monomer and multimer data in the following analysis.

**Scaling monomer data improves reconstruction for multimer.** Table 9 presents the reconstruction performance of the structure tokenizer. We observe that increasing the monomer data from 200K to 2M leads to a substantial improvement in both reconstruction RMSD and TM-score on the PDB-Multimer validation set. This indicates that monomer modeling is closely related to multimer modeling and could provide direct benefits to it.

**Chain linker and position offset.** We attempt to identify and bridge the gap between multimer and monomer data in Table 10. Noting that chains are typically spaced farther

*Table 10.* **Applying chain linker and position offset in multimer modeling.** We present the folding results on PDB-Multimer and report the reconstruction performance of structure tokenizer. ESM-Fold used G-linker of length 25 by default in multimer folding.

| Method | PDB-Multimer | |
|---|---|---|
| | RMSD↓ | TMscore↑ |
| *Tokenizer Reconstruction* | | |
| DPLM-2 (monomer) tokenizer | 6.873 | 0.784 |
| *w/ Pos. Offset* | 5.886 | 0.812 |
| *Folding* | | |
| ESMFold | 17.297 | 0.850 |
| DPLM-2 | 19.110 | 0.768 |
| *w/ Chain Linker* | **17.966** | 0.771 |
| *w/ Pos. Offset* | 18.338 | 0.767 |

*Table 11.* **Fine-tuning with multimer and monomer data.** We evaluate the effects of fine-tuning with PDB-Multimer and Swissprot on structure prediction. Incorporating multimer data improves both monomer and multimer folding. SFT: supervised fine-tuning with folding objective.

| Training Data | | SFT | PDB-Multimer | | CAMEO 2022 | |
|---|---|---|---|---|---|---|
| PDB -Multimer | Swissprot | | RMSD↓ | TMscore↑ | RMSD↓ | TMscore↑ |
| | √ | | 17.966 | 0.771 | 7.703 | 0.793 |
| | √ | √ | 19.615 | **0.799** | 6.612 | 0.823 |
| √ | | √ | **16.146** | 0.775 | 10.989 | 0.686 |
| √ | √ | √ | 16.674 | 0.798 | **6.410** | **0.831** |

apart than individual connecting residues, we apply a position index offset to each residue, which is calculated as the product of chain index and a predefined offset value. The offset is incorporated into the relative position embedding of the structure detokenizer (Wang et al., 2024a). We further examine the effects of connecting chains using glycine (G) linkers of varying lengths under the folding scenario. These linkers not only introduce a position offset but also serve as pseudo-connectors between protein chains. Our findings suggest that chain linkers and position offsets both improve the metrics. These results highlight the difference between multimer and monomer and suggest that properly differentiating chains in sequence and positional space are essential for effective multimer modeling.

**Finetuning on PDB-Multimer.** We study the effects of multimer and monomer data by fine-tuning DPLM-2 in Table 11. We excluded multi-chain proteins with lengths outside the range of $[60, 512]$, resulting in 3462 training samples in PDB-Multimer. We observe that incorporating PDB-Multimer into the training data effectively improves the structure folding for both multimer and monomer. This highlights that multimer data might be essential for robust structural modeling. Additionally, fine-tuning with multimer data is essential for reducing RMSD on PDB-Multimer. Since multimer chains are typically more spatially separated than neighboring residues in monomers, models trained solely on monomer data tend to yield a higher RMSD, which captures local atomic deviations. Meanwhile, the

monomer Swissprot data (200K) helps learn the high-level 3D structural patterns due to its larger dataset size compared to PDB-Multimer (3.5K), as reflected by consistently higher TMscores that measure global structural similarity.

# 7 Conclusions

In this work, we identify the limitations in structural modeling for multimodal protein language models and propose an effective design space to bridge the gap. We demonstrate that tokenization quantization loss can be effectively mitigated with bit-label supervision and flow-matching, which significantly improve the structure prediction accuracy. We introduce geometric inductive biases through architectural design and leverage representation learning to refine generation diversity. Building on the strengths of each component, we further investigate their orthogonality, which informs the final recommended setting. Lastly, to tackle the scarcity of structure data, we explore the data coverage to include multimers, ensuring broader 3D structural understanding. Our results show that these effective designs allow multimodal models to achieve on-par or even superior folding accuracy compared to larger, specialized folding models. Despite these, we also notice that there remain several limitations and future work directions deserving to be explored. We provide discussions on the **limitations** and **potentials** in §B. We believe this work will contribute to advancing the development of more effective multimodal protein language models.

## Acknowledgements

We thank anonymous reviewers for their valuable feedback. We would like to especially thank Dr. Hang Li for insightful discussions on the project and feedback on the manuscript that help shape this study, as well as Yi Zhou, Jing Yuan and Yilai Li for their valuable comments.

## Impact Statement

This paper presents work whose goal is to advance the field of machine learning and its applications on protein modeling. There are many potential societal consequences of our work. Our work on protein generation and representation learning can be used in developing potent therapeutic macromolecues such as antibodies and accelerate the research process of drug discovery. Our method may be adapted to other scenarios of computer-aided design, such as small molecule design, material design, and chip design. It is also needed to ensure the responsible use of our method and refrain from using it for harmful purposes.

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

# A    Taxonomy of the Design Space

We present the taxonomy of our design methods in Table 12, summarizing the corresponding space of each design choice with discussion on the underlying motivation. The findings from each design method are presented in the rightmost column.

*Table 12.* The taxonomy of design choices, their orthogonality, and a systematic synthesis of findings.

| Design Space | Design Choice | Traditional Choice | Motivation | Findings |
|---|---|---|---|---|
| Improved generative modeling | Bit-based modeling | Index-based modeling | Intuition: Small changes at the bit level can result in drastically different indices, making the learning process of index-based labels challenging.

Empirical Analysis: Direct index prediction is highly inaccurate (8.64% on Cameo). | Bit-level supervision improves prediction accuracy across index and bit-level, while substantially reducing structural deviation from ground truth (RMSD and TMscore). |
| | ResDiff | Decode tokens without residuals | Quantizing continuous tokens into discrete ones amplifies reconstruction errors, suggesting that recovering the lost residuals might enhance structure prediction accuracy. | ResDiff module performs fine-grained refinements on the local structure (Fig. 7), showing consistent performance improvements across different DPLM-2 variants. |
| | Hybrid approach—data-space sampling and predicting discrete tokens | Predict discrete tokens | Discrete structure tokenization inevitably sacrifices atomic-level details. | A hybrid approach with flow matching improves structure generation for the folding task and speeds up the sampling process by requiring 10x fewer sampling steps. |
| Structure-aware approaches | Geometry-aware architecture | Sequence-based transformer architecture | Protein structures require capturing higher-order relationships between residues, as evidenced in folding. | Geometric modules enhance structure folding performance and generation diversity. Unlike typical folding models, our analysis shows that triangle layers provide little benefit while greatly slowing down training. |
| | Representation alignments to folding model | Discrete token supervision only | Discrete token supervision might be less effective for capturing finer atom-level details. Unlike discrete supervision that enforces strict and sharp targets, representation alignment enables smooth, informative and high-dimensional learning, preserving finer structural nuances and improving generation diversity. | Representation alignment (Yu et al., 2024) considerably boosts generation diversity and is compatible with both language model-based and geometric architectures.

Alternative approaches:
1. Contact map supervision (esm2): The quadratic complexity $O(L^2)$ results in extreme inefficiency in our prelim. study.
2. Direct coordinate supervision: Need to carefully take into account se3 symmetry. Recent progress on this: Proteina (Geffner et al., 2025). |
| Data | Multimer data exploration | Monomer data only | Multimer data presents diverse structural arrangements, while most existing PLMs are trained solely on monomer. | Multimer and monomer modeling are deeply interconnected and leveraging multimer data advances structural modeling for both single and multi-chain proteins. |

# B    Discussions and Limitations

While our approach significantly improves structural modeling in multimodal protein language models, several limitations remain.

- First, despite advancements in structural tokenization and generative modeling, our method still relies on discrete representations of 3D structures, which inherently introduce information loss. While bitwise discrete modeling mitigates some quantization errors, it does not fully capture continuous geometric variations, limiting the fidelity of fine-grained structural features. Future research could explore hybrid approaches that combine discrete and continuous representations to enhance structural expressiveness. Additionally, atomic-level precision remains a fundamental challenge, as current tokenized representations primarily operate at residue or backbone levels, missing finer atomic interactions crucial for protein folding and function. Bridging the gap between token-based modeling and direct atomic-coordinate generation is essential for achieving true atomic-resolution structure modeling.
- Second, our framework, like other multimodal PLMs, lacks explicit physical constraints and energy-based priors, which are crucial for generating physically plausible protein structures. While representation-level learning helps refine structural understanding, incorporating differentiable physics-based priors or energy functions could further improve structural realism and biological validity.
- Third, our analysis is conducted on relatively small protein language models (up to 3B), and the scalability of these design choices remains uncertain. We chose this setup to prioritize the analysis of our design space. Further study into how these designs scale with larger models could further enhance multimodal protein language modeling, especially given the proven benefits of scaling in DPLM-2.
- Finally, our training dataset, while diverse, remains constrained by available high-quality structural data, particularly for multimers. While we demonstrate that leveraging multimeric structures enhances modeling, the sparsity of curated multimer datasets poses challenges for generalization. Future efforts should prioritize data augmentation techniques and larger-scale multimodal datasets to improve robustness across different protein families.

Ultimately, the future of multimodal protein foundation models should enable direct atom-level data-space structure modeling; however, this remains an unsolved challenge. Token-based multimodal protein LMs provide a scalable, practical, and effective approach to generative protein modeling. Advancing this approach while addressing its limitations—such as structural information loss, lack of physical constraints, and limited multimodal generalization—will be key to unlocking their full potential and provide crucial insights that help us move closer to achieving atomic-resolution protein modeling in a unified multimodal framework.

## B.1    Discussion on Alternative Approaches

We recap our methods to address O1-O3 (see Section 2.2) and discuss potential alternative approaches below.

**O1: Structure tokenization results in information loss.**

- Our approach: We train an additional continuous diffusion module ResDiff to recover the lost residuals of discrete tokens.
- Alternatives: Training DPLM-2 on continuous tokens. However, it's unclear whether this approach would work for joint modeling of structures and sequence, which is inherently discrete.

**O2: High reconstruction accuracy of structure tokenizer does not ensure better structural generative results.**

- Our approach: We primarily improve the designs of language modeling, including structure-aware generative modeling, architectural designs, and representation alignments.
- Alternatives: Adopt direct modeling in the data space following a similar manner as Proteina (Geffner et al., 2025). Similarly, it's unclear if direct modeling is robust for joint modeling of discrete sequences and continuous 3D structures.

**O3: Multimodal PLM gets index-based structure tokens miserably wrong.** Small bit-wise changes could result in a dramatically different index label. This challenge intensifies as codebook size grows, making direct index prediction even more difficult.

- Our approach: Finer-grained token prediction might resolve such challenge. We achieve a finer-grain prediction on the "dimension-level" using bit-based labels in contrast to index-based labels.
- Alternatives: Another potential direction is to hierarchically tokenize the structures to achieve fine-grained tokens at the "resolution" level following methods like RQ-VQVAE (Lee et al., 2022a) and VAR (Tian et al., 2024). However, it remains elusive how to well define "resolution" of proteins as a natural choice.

# C   Implementation Details

## C.1   Residual Diffusion Module

We use another light-weight diffusion module on the top of DPLM-2 to predict the residuals information $\mathbf{z}_{\text{cont}} - \mathbf{z}_{\text{quant}}$, conditioned on the language model output discrete tokens $\mathbf{z}_{\text{quant}}$ and hidden states $\mathbf{h}$. We employ linear projection on the discrete structure tokens and hidden states of each layer to obtain the condition information $\mathbf{c}$:

$$\mathbf{c} = \mathbf{z}_{\text{quant}} W_{\text{quant}} + \sum_i^N a_i h_i,$$

where $N$ represents the number of layers and $a_i$ represents the weight of hidden state of layer $i$, which is obtained by a learnable vector $\mathbf{w} = (w_1, w_2, ..., w_N) \in \mathbb{R}^N$ and $a_i = \frac{e^{w_i}}{\sum_j^N e^{w_j}}$. The condition information is subsequently processed by an adaptive layer norm block, similar to Peebles & Xie (2023). During generation, the language model first predicts the discrete structure tokens $\mathbf{z}_{\text{quant}}$ while providing the hidden states of each layer $\mathbf{h}$, then the residual diffusion module takes the $\mathbf{z}_{\text{quant}}$ and $\mathbf{h}$ as conditions and generates the residuals information $\mathbf{r}$. Finally, we add the residuals and the discrete structure tokens to obtain the continuous structure feature $\mathbf{z}_{\text{cont}}$, which is decoded to 3D protein structure.

The light-weight diffusion module consists of 6 layers of MLP with a hidden size of 1024. During training, the learning rate is warmed up over the first 2,000 steps to a peak value of $1 \times 10^{-4}$ and then linearly decayed to $1 \times 10^{-5}$. We train the residual diffusion module for 100,000 steps with a batch size of 240.

## C.2   Bit-level Supervision

As shown in Figure 1, we explore the bit-level supervision instead of index-level supervision. The language model takes the K-dimension quantized feature as input, and leverage $K$ binary classifiers to predict each bit of the structure feature in parallel, considering that the each bit of the quantized feature is either $-1$ or $+1$. For the input quantized structure feature $\mathbf{z}_{\text{quant}} \in \{-1, +1\}^{L \times K}$, we employ a linear projection $W_{\text{input}} \in \mathbb{R}^{K \times H}$, where $H$ represents the hidden size of language model, to obtain the input embedding for the transformer block. We utilize another linear projection $W_{\text{output}} \in \mathbb{R}^{H \times 2K}$ to obtain the logits of the output quantized feature. During sampling, the logits with dimension (L, 2K) are reshaped to (L, K, 2), and we take the softmax operation at the last dimension to get the probability of each bit. The bit-level training loss is calculated on each bit by binary cross-entropy. Considering that the number of index-level vocabulary is $2^K$ and if we use index-level supervision we would need to train the input and output projection with $2^K * H$ parameters, the bit-level supervision only needs to train the input and output projection with $K * H$ parameters, significantly reducing the training cost.

In the training stage, we follow Wang et al. (2024a) and utilize the pre-trained DPLM (Wang et al., 2024b) as the parameter initialization to inherit the evolutionary representations learned from the massive protein sequence data, which implicitly captures the structural information that benefits structure modeling (Lin et al., 2022). In addition to the parameters inherited from DPLM, we randomly initialize the structure input-output linear layers. The bit-level DPLM-2 is grounded in the *absorbing* discrete diffusion framework (Austin et al., 2021; Zheng et al., 2023a). To adapt to the absorbing discrete diffusion framework, we also introduce a learnable absorbing embedding for the absorbing state of structure. We employ 2,000 warmup steps until reaching the maximum learning rate $1 \times 10^{-4}$, and utilize a linear decay scheduler to decay LR to $1 \times 10^{-5}$. The overall training process consists of 300,000 steps.

## C.3   Hybrid Approach for Direct Data-space Sampling

We fine-tune the bit-based DPLM-2 to denoise continuous latent structural features, encoder($\mathbf{x}_t$), following a strategy similar to flow matching (FM). The structure encoder, DPLM-2, and structure decoder together form our structure denoising pipeline. During training, DPLM-2 predicts denoised quantized features $\mathbf{z}_{\text{denoised}}$, which are decoded into structures $\bar{\mathbf{x}}_{\text{denoised}}$ at inference time. Instead of using the FM loss, we employ a bit-wise cross-entropy loss between $\mathbf{z}_{\text{denoised}}$ and the ground truth bit labels $\mathbf{z}_{\text{quant}}$, avoiding the computational overhead of running the structure decoder during training.

## C.4   Geometric Designs

As shown in Figure 2, we integrate geometry-aware modules into the encoder blocks of PLMs. To initialize the pair representations, we follow Multiflow (Campbell et al., 2024b) to cross concatenate the input hidden representation with dimension (L, D) into a 2D representation map with dimension (L, L, D), followed by a 3-layer MLP. We use L and D to denote the length and feature dimension. The feature dimension D is selected as 1280 for structure representations and 128 for pair representations. We adopt the encoder block of PairFormer (Abramson et al., 2024) as the structure attention module in Figure 3. The hidden dimension in triangle update operations is selected as 128, while the dimension of hidden pair representation in triangle attention operations is 32. We use 4 and 16 heads in the triangle attention and

attention-with-pair-bias modules respectively. Transition layers consist of gated mechanisms and a 2-layer MLP which expands the dimension of hidden representations by a ratio of 4. We select the hidden dimension as 320 and use 16 heads in the attention-with-pair-bias module of the `SeqStruct` attention module in Figure 4.

### C.5 Representation Alignment

The process of REPA has been discussed in Section 4.2, we provide additional details below.

**Overview.** Representation alignment (Yu et al., 2024) is originally proposed in vision tasks which aligns the hidden embeddings with meaningful representations from the self-supervised pretrained encoders like DINOv2 (Oquab et al., 2023), dramatically improving the visual generation quality the the training efficiency of diffusion models (Rombach et al., 2022). Here we extend this technique to protein modeling by aligning representations from protein language models with representations from the folding model. Formally, we define the folding model as $F = f_2 \circ f_1$, where $F(\mathbf{s}) = \mathbf{x}$, with $\mathbf{s}$ denoting the sequence and $\mathbf{x}$ denoting the structures. The model first computes the semantic embeddings $f_1(\mathbf{s}) = \mathbf{y} \in \mathbb{R}^{L \times D}$, which encode rich 3D structural information that allows for precise structure predictions. Additionally, we extract the hidden representations $f_\theta(\mathbf{s}^*, \mathbf{x}^*) = \mathbf{h}$ from our protein language model, where $\mathbf{s}^*$ and $\mathbf{x}^*$ represent sequence and structure tokens with masked entries. REPA aligns $h_\phi(\mathbf{h}) \in \mathbb{R}^{L \times D}$ with the target representation $\mathbf{y}$, where $h_\phi(\mathbf{h})$ is a trainable 3-layer MLP that ensures the alignment of feature dimension. This alignment is achieved by maximizing the similarity between two representations:

$$\mathcal{L}_{REPA}(\theta, \phi) = -\frac{1}{L} \sum_{i=1}^{L} \mathtt{sim}(y^{[i]}, h^{[i]}),$$

where $i$ is the residux index, and $\mathtt{sim}(\cdot, \cdot)$ is the cosine similarity function.

**Precompute pair and structure representations.** Using the protein sequences in our training data, we precompute the structure and pair representations of the folding model. We select ESMFold (Lin et al., 2023) as the target encoder due to its computational efficiency and open-source availability in comparison to other folding models (Jumper et al., 2021; Abramson et al., 2024). In particular, we extract the output representations of the folding trunk in ESMFold. We run three iterations of recycling mechanisms within the folding trunk.

**Multi-layer ensemble.** Instead of aligning embeddings from a specific layer as in (Yu et al., 2024), we apply a learnable `nn.Parameter`, followed by a softmax function, to weight and aggregate representations across all layers. We refer to this approach as multi-layer ensemble. We use multi-layer ensemble as it achieves slightly better performance in folding RMSD and more importantly, eliminates the need to select the alignment layer as a hyperparameter.

### C.6 Folding Supervised Finetuning

The folding SFT is to supervise the DPLM-2 to generate structure tokens given sequence. During training, we keep the sequence tokens unmasked and only mask the structure tokens. In Tables 4 and 7, we perform folding SFT based on the pretrained model to further enhance folding performance.

## D More Empirical Analysis

### D.1 Structure-aware Generation and Predictive Tasks

**Inverse folding (structure-conditioned sequence generation).** We report the amino acid recovery (AAR) self-consistency TMscore on the Cameo dataset in Table 13. Our results show that the 650M-parameter DPLM-2 variant, with bit-based modeling, outperforms the 3B-parameter baseline. Incorporating geometric architectural designs and representation alignments leads to further improvements. These findings indicate that these design methods contribute positively beyond structure generation.

*Table 13.* **Structure-conditioned sequence generation**.

|  | CAMEO 2022 | |
| --- | --- | --- |
|  | AAR↑ | TMSCORE↑ |
| DPLM2 650M | 0.4962 | 0.8816 |
| DPLM2 3B | 0.5236 | 0.8900 |
| DPLM2 BITWISE | 0.5586 | 0.8907 |
| GEO + BITWISE | 0.5665 | 0.8886 |
| GEO + BITWISE + REPA | **0.5681** | **0.8909** |

**Structure-aware protein representation learning.** In Table 14, we evaluate the performance of the models in learning structure-aware protein representations, following the evaluation protocol of SaProt (Su et al., 2023). Bit-based supervision

further improves the representation learning performance of DPLM-2. We attribute this to the finer granularity of bitwise labels compared to conventional index-based ones, allowing the multimodal protein language model to capture the structural semantics in latent space more effectively.

Table 14. **Structure-aware representation learning.**

| MODEL | HUMANPPI | DEEPLOC SUBCELLULAR |
|---|---|---|
| | ACC (%) | ACC (%) |
| SAPROT | 86.41 | **85.57** |
| DPLM-2 | 84.44 | 82.98 |
| DPLM-2 BITWISE | **88.89** | 83.39 |

## D.2  Case Study of Residual Diffusion

We present a visualization of the protein structures generated by the protein language model (PLM) and their refined counterparts through residual diffusion, with structural alignment shown in Figure 7. In this visualization, gray structures represent the initial PLM predictions, while blue structures denote the residual diffusion refinements. The dark blue regions highlight specific areas where residual diffusion has effectively adjusted the local structural details, as reflected in the secondary structure transition from flexible loops to beta strands. These visualizations demonstrate residual diffusion's capability to supplement fine-grained structural variations, rectify the potential local prediction errors from the PLM, and facilitate the formation of plausible secondary structures.

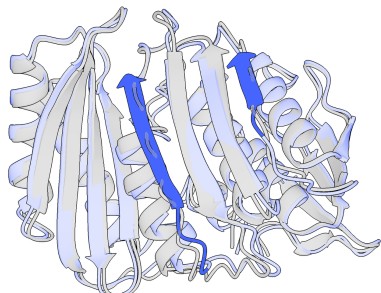

Figure 7. **Case study of residual diffusion.** We visualize the difference between the PLM generated structure and residual diffusion refined structure. The gray and blue structure correspond to the PLM prediction and the residual diffusion-refined one, respectively.

## D.3  Representaion Alignment

Table 15. **REPA alignment target.** We analyze the impact of aligning different representations on folding performance and generation diversity using GeoDPLM with 150M parameters. All models are trained for 40k steps. Aligning structural representations improve both tasks.

| TARGET | | CAMEO 2022 | | DIVERSITY |
|---|---|---|---|---|
| STRUCT | PAIR | RMSD↓ | TMSCORE↑ | CLUSTERS↑ |
| × | × | 9.410 | 0.757 | 0.563 |
| ✓ | | **9.069** | **0.766** | 0.756 |
| | ✓ | 9.467 | 0.764 | **0.769** |
| ✓ | ✓ | 9.448 | 0.761 | 0.700 |

**Alignment target ablation.** We use GeoDPLM with 150M parameters to study the effects of aligning different types of representations in Table 15. TMscores improves across all settings, suggesting that representation alignment guides the model to capture high-level 3D structural semantics. Aligning pair representations notably refines the generation diversity but slightly degrades the folding RMSD, which emphasizes precise local atom positions. Additionally, incorporating pair representations intensifies the I/O bottleneck, slowing training by approximately five times compared to using only structure representations. Aligning structural representations improve both the generation diversity and folding tasks, while ensuring a similar training efficiency as the baseline.

**Ensemble weight distribution of multi-layer ensemble.**  We present the learned ensemble weight distribution across

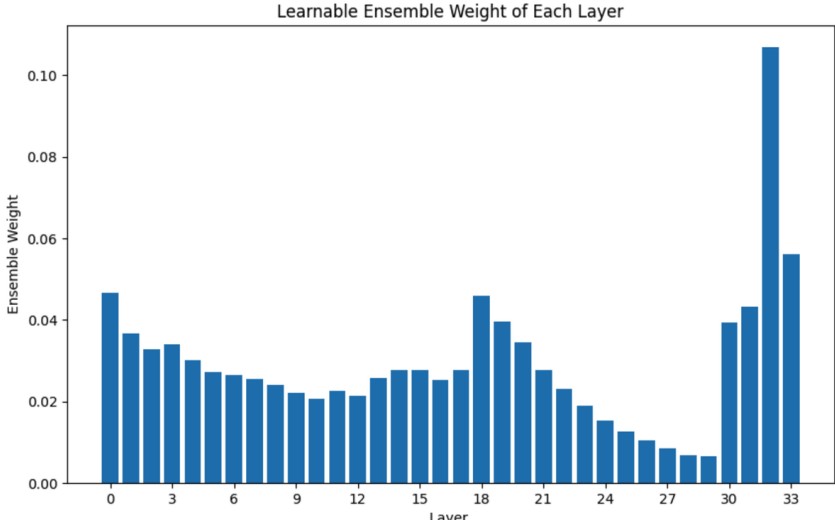

*Figure 8.* **Ensemble weight distribution across layers.** We visualize the learned ensemble weight after applying the softmax function. By leveraging multi-layer ensemble, we avoid the need for manually selecting the alignment depth as a hyperparameter in REPA.

layers in Figure 8. The weights are mostly concentrated in the last four layers, with the highest value occurring at the second-to-last layer. However, the value (0.1069) is not significantly dominant over other values.

### D.4 Multimer Exploration

**Chain linker and position index offset.** We present a more comprehensive study by varying the lengths of chain linkers and position offset values in Tables 16 and 17. As shown in Table 16, the reconstruction accuracy steadily improves as the offset increases from 0 to 5, and eventually reaching saturation at 25. In Table 17, chain linkers provide a similar benefit to position offsets, with folding RMSD improving as the linker length increases, reaching optimal performance at a length of 25.

*Table 16.* **Position offset improves multimer reconstruction.** We apply an offset to the position indices in the relative position embedding of the structure detokenizer to differentiate between chains, where the offset is determined by the product of chain index and a predefined value.

| OFFSET VALUE | PDB-MULTIMER | |
|:---:|:---:|:---:|
| | RMSD↓ | TMSCORE↑ |
| 0 | 6.873 | 0.784 |
| 1 | 6.498 | 0.790 |
| 3 | 6.112 | 0.799 |
| 5 | 5.967 | 0.805 |
| 10 | 5.937 | 0.811 |
| 25 | **5.886** | **0.812** |
| 50 | 5.891 | 0.812 |

*Table 17.* **Apply chain linker and position offset in multimer structure prediction.** We use glycine (G) of various lengths as the linker to connect different chains, and employ position offset in the position indices within both DPLM-2 and the detokenizer during structure folding. ESMFold used G-linker of length 25 by default in multimer folding.

|  | Length/Value | PDB-Multimer | |
| --- | --- | --- | --- |
|  |  | RMSD↓ | TMscore↑ |
| ESMFold | 25 | 17.297 | 0.850 |
| DPLM-2 | 0 | 19.110 | 0.768 |
| *w Chain Linker* | 1 | 19.044 | 0.770 |
|  | 3 | 18.757 | 0.770 |
|  | 5 | 18.579 | **0.773** |
|  | 10 | 18.808 | 0.768 |
|  | 25 | **17.966** | 0.771 |
|  | 50 | 18.208 | 0.770 |
| *w Pos. Offset* | 1 | 18.848 | 0.769 |
|  | 5 | 18.700 | 0.770 |
|  | 10 | 18.338 | 0.767 |
|  | 25 | 18.608 | 0.770 |

### D.5 Analysis on Pretraining

**Bit-based pretraining for hybrid modeling.** As shown in Table 18, the "bitwise & FM" variant initialized from the bit-based DPLM-2 outperforms the model trained from scratch across both datasets. This demonstrates the potential of using bit-based DPLM-2 initialization to support hybrid generative modeling. This benefit might be attributed to: (1) DPLM-2 benefits from sequence pretraining that improves structural modeling (Lin et al., 2022); (2) Sequence data vastly outnumbers experimental structures, making pretraining more effective.

*Table 18.* **Effects of pretraining on hybrid modeling**.

|  | Cameo2022 | | PDB date | |
| --- | --- | --- | --- | --- |
|  | RMSD | TMScore | RMSD | TMScore |
| Bitwise & FM *init. from DPLM-2* | 6.1825 | 0.8414 | 2.8697 | 0.9099 |
| Bitwise & FM *training from scratch* | 10.9815 | 0.7090 | 6.4453 | 0.8070 |

### D.6 Training and Sampling Efficiency

**Bit-based vs. hybrid approach.** We report the training time of DPLM-2 variants with either bit-based modeling or hybrid approach on 16 H100s for 300k training steps in Table 19. The increased training time of flow matching (FM) primarily results from the on-the-fly computation of noisy structure $x_t$ using structure encoder, as we can precompute the discrete structure tokens when FM is not applied. FM remains highly effective when inference efficiency is a priority, as it accelerates the sampling process by 10x by requiring fewer sampling steps.

*Table 19.* **Efficiency comparison.** We compare the training and sampling efficiency of the DPLM2 variants using either bit-based or hybrid modeling.

|  | # SAMPLING STEPS | TRAINING TIME (300K STEPS) |
| --- | --- | --- |
| *w/o FM* | 100 | 46 HRS |
| *w/ FM* | 10 | 81 HRS |

## E Related Work

**Protein language models.** Impacted by the success of large language models (LLMs), similar practice has been extended to the development of PLMs. For instance, sequence-based protein language models operate by taking each amino acid sequence as a sentence. ESM-1b (Rives et al., 2019) utilizes self-supervised masked language modeling on 250 million protein sequences spanning evolutionary diversity, later leading to the development of ESM-2 (Lin et al., 2023) that further extends the scaling laws. DPLM (Wang et al., 2024b) adopts the bi-directional receptive field and scalable discrete diffusion

pre-training process (Ye et al., 2023b;a) which allows DPLM to exhibit a strong generative capability. ProtTrans (Elnaggar et al., 2021), ProteinBERT (Brandes et al., 2022), PRoBERTa (Nambiar et al., 2020), ProtAlbert (Behjati et al., 2022), TAPE (Rao et al., 2019), ProteinLM (Xiao et al., 2021), and CARP (Yang et al., 2022) involve several other representative masked language modeling (MLM) paradigm. These sequence-based PLMs perform competitively with classic methods that rely on multiple sequence alignments, indicating that PLMs have captured some of the evolutionary information from sequences alone. In particular, these protein language models achieve powerful generalization on various downstream tasks involving the secondary and tertiary structures. Recent findings further showcase their capabilities in predicting protein functions (Meier et al., 2021), structure folding (Lin et al., 2023), and de novo designs (Verkuil et al., 2022).

**Multimodal protein language models.** Proteins are defined by amino acid sequences and three-dimensional structures, where the sequence dictates the structure, and the structure, in turn, determines the protein's function. Due to this intrinsic relationship, there has been increasing interest in developing multimodal PLMs that integrate both sequence and structural information. Many recent research has explored structure-enhanced PLMs, building upon the success of sequence-based PLMs (Zheng et al., 2023b). The key foundation of this direction is the adoption of VQ-VAE (Van Den Oord et al., 2017b), which encodes the 3D structures into discrete tokens that represent the geometric conformation of each protein residue. For instance, SaProt (Su et al., 2023) constructs a structure-aware vocabulary that integrates both 3D and amino acid tokens. ProSST (Li et al., 2024) models the correlations between amino acids and structure tokens through a disentangled attention mechanism. ProstT5 (Heinzinger et al., 2023) extends the sequence-based ProT5 (Elnaggar et al., 2021) by training it to predict structure and sequence tokens conditionally. ESM-3 (Hayes et al., 2024) treats sequence and structures as separate tracks and applies an all-to-all masked language modeling approach. DPLM-2 (Wang et al., 2024a) further advances the joint distribution of sequence and structures through a discrete diffusion mechanism, enabling it to effectively denoise the masked sequence or structure tokens. This method allows DPLM-2 to learn meaningful representations that enhance predictive tasks, while achieving strong generative capabilities, such as conditional sequence generation and co-generation of sequence and structures.

**Protein structure prediction.** Leveraging deep learning approaches, research in structure folding and structure generation has advanced significantly. One key direction is end-to-end structure prediction, where models map amino acid sequences to three-dimensional structures. Notable breakthroughs include the AlphaFold series (Senior et al., 2020; Jumper et al., 2021), ESMFold (Lin et al., 2023) and RoseTTAFold (Baek et al., 2021). As structure folding does not require sequence generation, these methods typically operate directly in the original data space, as seen in AlphaFold3 (Abramson et al., 2024). In addition, many of these methods innovate architectures by incorporating geometric properties of proteins. For example, OmegaFold (Wu et al., 2022) introduces Geoformer, which iteratively refines the sequence and pairwise representations to minimize the geometric inconsistencies. AlphaFold2 (Jumper et al., 2021) employs Evoformer to model 2D pairwise representations and update them through a series of triangle operations. These designs highlight the importance of capturing higher-order relations for accurate structure prediction.

**Protein structure generation.** Another growing line of work involves structure generation, which aim to generate novel protein structures and functional design (Ye et al., 2025). In this direction, diffusion-based approaches have demonstrated remarkable success. For example, RFDiffusion (Watson et al., 2023) excels at guiding protein structure designs to meet specific functional constraints like enzymes. Chroma (Ingraham et al., 2023) enables the generation of proteins and protein complexes and allows flexible property-based prompting. This approach efficiently generates large structures by transforming collapsed polymers into protein backbones and sequences. ProteinSGM (Lee et al., 2022b) facilitates novel 3D protein design using 2D maps of distances and angles as conditions. Multiflow (Campbell et al., 2024b) achieves structure-sequence co-generation by developing multimodal flow matching that allows the hybrid modeling of continuous structural data and discrete sequences. Despite the promising success in structure generation, these approaches often struggle in sequence-conditioned structure prediction, primarily due to the lack of large-scale training on sequence databases. Recent work further includes sidechains and extend to modeling all-atom structures for generating protein complexes and function design (Chen et al., 2025).

