# OpenReview forum: "Elucidating the Design Space of Multimodal Protein Language Models"
_ICML.cc/2025/Conference — ICML 2025 spotlightposter_

### Official Review · Reviewer_RSLA · 2025-03-12

**Overall Recommendation:** 4

**Summary:**

The paper discusses the design space of multimodal protein language models through the lens of discretizing protein structures. The paper attempts to address a key challenge with information loss of fine-grained structural details upon tokenization. Multimer structures are considered, and BIT-based modeling is used to improve structural tokenization and generation from multimodal models.

**Claims And Evidence:**

Yes

**Essential References Not Discussed:**

May be of interest to the authors:

Lu, Amy X., et al. "Compressing the Latent Space of Single-Sequence Protein Predictors for Multimodal Generation." ICML 2024 Workshop on Efficient and Accessible Foundation Models for Biological Discovery.

**Experimental Designs Or Analyses:**

Yes

**Methods And Evaluation Criteria:**

Yes

**Other Comments Or Suggestions:**

See above

**Other Strengths And Weaknesses:**

Considering representations of multimeric systems is an important problem in protein design and progress in this area could have significant downstream applications. Addressing limitations in current structure tokenization methods is similarly important and well-motivated.

Can the authors comment on any simple baselines or alternative approaches they pursued to resolve the identified problems (O1, O2, and O3)?

The paper discusses SaProt but does not explicitly compare to it or other related "structure-aware" pLMs beyond ESM3. Can the authors comment on their reasoning for including/not including specific comparisons?

**Questions For Authors:**

See above

**Relation To Broader Scientific Literature:**

Well contextualized and motivated with respect to challenges in the field, specifically MultiFlow and ESM3.

**Theoretical Claims:**

N/A

---

> ### Author Rebuttal · Authors · 2025-04-01
>
> Many thanks for your comments that have greatly improved our manuscript! We address your concerns as below. We sincerely thank you once again and welcome any further feedback!
>
> > Q1: Can the authors comment on any simple baselines or alternative approaches they pursued to resolve the identified problems (O1, O2, and O3)?
>
> We recap our approaches to address O1-O3 and discuss potential alternative approaches below.
>
> **O1: Structure tokenization results in information loss**
>
> **Our approach:** We train an additional continuous diffusion module ResDiff to recover the lost residuals of discrete tokens.
>
> **Alternatives:** Training DPLM2 on continuous tokens. However, it's unclear whether this approach would work for joint modeling of structures and sequence, which is inherently discrete.
>
> **O2: High reconstruction accuracy of structure tokenizer does not ensure better structural generative results.**
>
> **Our approach:** We primarily improve the designs of latent structure modeling, including structure-aware generative modeling, architectural designs, and representation alignments.
>
> **Alternatives:** Adopt direct modeling in the data space following a similar manner as Proteina [1]. Similarly, it's unclear if direct modeling is robust for joint modeling of discrete sequences and continuous 3D structures.
>
> **O3: Multimodal PLM gets index-based structure token prediction miserably wrong.**
> Small bit-wise changes could result in a dramatically different index label. This challenge intensifies as codebook size grows, making direct index prediction even more difficult.
>
> **Our approach:** Finer-grained token prediction might resolve such challenge. We achieve a finer-grain prediction on the "dimension-level" using bit-based labels in contrast to index-based labels.
>
> **Alternatives:** Another potential direction is to hierarchically tokenize the structures to achieve fine-grained tokens at the "resolution" level following methods like RQ-VQVAE [2] and VAR [3]. However, it remains elusive how to well define "resolution" of proteins as a natural choice.
>
> > Q2: The paper discusses SaProt but does not explicitly compare to it or other related "structure-aware" pLMs beyond ESM3. Can the authors comment on their reasoning for including/not including specific comparisons?
>
> Thanks for your question. In our submitted version, we focus on the generative capabilities of protein structure (e.g., folding). However, SaProt focuses on the protein representation learning ability, which utilizes structure tokens as the representation for structure input and is not capable of generating structure.
>
> In the table below, we also provide the results of structure-aware protein representation learning, following the setting of SaProt. Due to the limited time, we only evaluate on the HumanPPI and DeepLoc Subcellular tasks, and we will provide results of more protein predictive tasks in the final version.
> As seen, the proposed bit-based modeling is able to further improve the representation learning performance of DPLM-2. We hypothesize that the bit-based modeling, which is finer-grained, is more suitable to learn and enables the multimodal PLM to capture the structural pattern in latent space more effectively, thus enhancing structure-aware representation learning.
>
> | Model | HumanPPI | DeepLoc Subcellular |
> |---|---|---|
> |  | Acc (%)↑ | Acc (%)↑ |
> | SaProt | 86.41 | 85.57 |
> | DPLM-2 | 84.44 | 82.98 |
> | DPLM-2 Bit | 88.89 | 83.39 |
>
> [1] Geffner et al. Proteina. ICLR 2025.
>
> [2] Lee et al. RQ-VAE. CVPR 2022.
>
> [3] Tian et al. VAR. NeurIPS 2024.
>
>
> > Essential References Not Discussed: Lu, Amy X., et al. "Compressing the Latent Space of Single-Sequence Protein Predictors for Multimodal Generation."
>
> Thanks for pointing this out to us, and we will elaborate on our discussion below in the final version.
>
> This work compresses the latent space of ESMFold to form a joint latent space of protein sequence and structure, and learn a continous generative model to sample compressed latent embeddings, which is then decoded by the ESMFold structure head and a learned sequence head. On the other hand, the multimodal protein language models like DPLM-2, use discrete diffusion generative framework to generate the discrete structure token, then decode using the learned structure detokenizer.
>
> The CHEAP embedding introduced by this work is an effective compression of ESMFold latent space. Integrating the CHEAP embedding with multimodal protein language models holds great potential for effective structural modeling. This is a very exciting direction and we will leave it for future work.

---

> > ### Comment · Reviewer_RSLA · 2025-04-03
> >
> > I appreciate the authors' response. As I recommended acceptance in my initial review, I will maintain that recommendation and score.

---

> > > ### Author Response · Authors · 2025-04-04
> > >
> > > Dear Reviewer RSLA,
> > >
> > > Thank you for taking time to carefully review our work and your thoughtful feedback. We sincerely appreicate your feedback and the efforts in engaging with our rebuttal, and we are grateful for the opportunity to address your concerns.
> > >
> > > Best,
> > >
> > > Authors

---

### Official Review · Reviewer_3QKf · 2025-03-13

**Overall Recommendation:** 4

**Summary:**

The manuscript systematically explores the design space of multimodal PLMs, aiming to identify and address their existing limitations. The authors highlight tokenization loss and inaccurate structure token predictions as major bottlenecks in structure prediction performance. To mitigate these issues, they propose multiple strategies, including bit-wise prediction, quantization residue modeling, and a hybrid generative approach that integrates the pretrained encoder-decoder with the PLM. The study is supported by comprehensive analyses and experiments, which empirically validate the conclusions and demonstrate notable performance improvements.

**Claims And Evidence:**

Yes

**Essential References Not Discussed:**

No

**Experimental Designs Or Analyses:**

Expriments are well designed and alalyses are logically reasonable.

**Methods And Evaluation Criteria:**

Yes

**Other Comments Or Suggestions:**

No other comments.

**Other Strengths And Weaknesses:**

Strengths:
The paper is well-written, with a clear structure and comprehensive experiments.

Weaknesses:

1. As mentioned in the supplementary material, the manuscript does not currently address structure-conditioned protein generation or structure-aware protein predictive tasks, which are crucial for functional protein design.

2. The performance on multimer data is particularly intriguing. When trained on SwissProt without PDB multimer training data, the TM-score is higher than when fine-tuned with PDB multimer data, yet the RMSD is worse. A more in-depth analysis of this phenomenon would provide valuable insights.

**Questions For Authors:**

Will the hybrid generative approach significantly increase training time? Providing detailed training time measurements would be helpful for assessing its feasibility and computational cost.

**Relation To Broader Scientific Literature:**

Multimodal protein language models have been shown in previous research to be highly effective for protein representation. However, their performance in structure prediction remains suboptimal. This paper makes key contributions by identifying the major limitations that restrict performance and proposing targeted strategies to overcome them, leading to significant improvements in folding prediction. Enhancing structure prediction not only increases modeling accuracy but also improves structure generation, ultimately advancing protein design.

**Theoretical Claims:**

The claims in the manuscript are supported by empirical results.

---

> ### Author Rebuttal · Authors · 2025-04-01
>
> Many thanks for your comments that have greatly improved our manuscript! We address your concerns as below. We sincerely thank you once again and welcome any further feedback!
> > W1: The current manuscript does not address structure-conditioned protein generation or structure-aware protein predictive tasks.
>
> Thanks for your suggestion. We provide additional results on structure-conditioned protein generation and protein predictive tasks below.
>
> **(1) Inverse Folding (structure-conditioned sequence generation)**.
>
> We report the amino acid recovery (AAR) and self-consistency TMscore on the Cameo dataset. Our results show that the 650M-parameter DPLM-2 variant, with bit-based modeling, outperforms the 3B-parameter baseline. Incorporating geometric architectural designs and representation alignments leads to further improvements. These findings indicate that these design methods contribute positively beyond structure generation.
>
> |  | Inverse Folding - Cameo 2022 |  |
> |---|---|---|
> |  | AAR↑ | TMscore↑ |
> | DPLM-2 650M | 0.4962 | 0.8816 |
> | DPLM-2 3B | 0.5236 | 0.8900 |
> | DPLM-2 Bit | 0.5586 | 0.8907 |
> | Geo + Bit | 0.5665 | 0.8886 |
> | Geo + Bit + REPA | **0.5681** | **0.8909** |
>
> **(2) Structure-aware protein representation learning.**
>
> In the table below, we provide the results of structure-aware protein representation learning, following the setting of SaProt. Due to the limited time, we only evaluate on the HumanPPI and DeepLoc Subcellular tasks, and we will provide results of more protein predictive tasks in the final version.
>
> As seen, the proposed bit-based modeling is able to further improve the representation learning performance of DPLM-2. We hypothesize that the bit-based modeling, which is finer-grained, is more suitable to learn and enables the multimodal PLM to capture the structural pattern in latent space more effectively, thus enhancing structure-aware representation learning.
>
> | Model | HumanPPI | DeepLoc Subcellular |
> |---|---|---|
> |  | Acc (%)↑ | Acc (%)↑ |
> | SaProt | 86.41 | 85.57 |
> | DPLM-2 | 84.44 | 82.98 |
> | DPLM-2 Bit | 88.89 | 83.39 |
>
> > W2: The model fine-tuned without PDB multimer data achieved a higher TM-score but a lower RMSD. A more in-depth analysis would provide valuable insights.
>
> Thanks for bringing up this discussion. RMSD captures local atom deviations, while TM-score measures global structural similarity. In multimers, chains are typically spaced farther apart than individual connecting residues, leading to a higher RMSD for models trained on only monomers. This highlights the importance of finetuning with multimer data to reduce RMSD.
> Meanwhile, fine-tuning on Swissprot (200K) helps learn the global protein structures better due to its larger dataset compared to PDB-Multimer (3.5K). As shown in Table 10 of our paper, incorporating Swissprot consistently achieves higher TM-score on both PDB-Multimer and Cameo.
>
> > Q1: Does the hybrid generative approach with flow matching significantly increase training time?
>
> We report the training time of the following DPLM-2 variants on 16 H100s for 300k training steps.
> The increased training time of flow matching (FM) primarily comes from the on-the-fly computation of noisy structure $\mathbf{x}_t$ using structure encoder, as we can precompute the discrete structure tokens when FM is not applied. FM remains highly effective when inference efficiency is a priority, as it accelerates the sampling process by 10x by requiring fewer sampling steps.
>
> |  | # Sampling steps | Training Time (300k steps) |
> |---|---|---|
> | w/o FM | 100 | 46 hrs |
> | w/ FM | 10 | 81 hrs |

---

### Official Review · Reviewer_i971 · 2025-03-13

**Overall Recommendation:** 4

**Summary:**

This paper performs an exploration of how to improve multimodal protein language models that jointly model both protein sequences and structures. The paper identifies limitations in the existing literature of token-based multimodal PLMs and propose (and explore) many design choices for such PLMs. The paper identifies two issues in token-based multimodal PLMs: (i) information loss when converting continuous 3D structures into discrete tokens and (ii) difficulty in accurate structure token prediction. They develop several design choices to address this and show that their design methods improve the structural modeling capabilities of multimodal PLMs.

**Claims And Evidence:**

The paper explores many design choices and evaluates them empirically. Most claims and design choices are given appropriate empirical evaluation which seems to be well supported. I'll provide three examples of this. First, bit-wise modeling seems to improve structure prediction accuracy as showcased in Table 3 and Table 4. Second, representation alignment does seem to enhance model performance (shown in Table 6).  Third, position offset and chain linkers help in multimer modeling (Table 9, Table 10).

The overall analysis is done on relatively small PLMs (up to 3B). I don't see this as an issue to the claims and I understand it's hard to scale up the model size if compute is limited, but it would be great to acknowledge more explicitly that the tests are done on smaller PLMs and that there is uncertainty in how it scales. This is especially true because there are no evaluations about how this performance and gains scale with the size of the PLMs.

**Essential References Not Discussed:**

N/A

**Experimental Designs Or Analyses:**

There are quite a few experimental designs in this paper (it's even a bit hard to follow all of them). Two main questions.

First, on structure tokenization analysis. The bit-level and index-level evaluation is really stark. How come? Is there any broader intuition behind this that you could check experimentally?

Second, the ablation study comparing different improved methods is nice and well presented. For some improvements (e.g. in Table 4), it's hard to know what's a genuine improvement because there are no uncertainty intervals (e.g. standard deviations for splits).

**Methods And Evaluation Criteria:**

The paper's methods for evaluating different design choices is nice and convincing. It provides limitations of existing token-based multimodal PLMs (especially information loss from structure tokenization). The proposed methods are quite broad, they target multiple aspects of the problem at once (tokenization, architecture, representation learning, data) with the experiments showing improvements from the design choices. Their methods primarily build on DPLM-2.

Given that this is a protein structure prediction task, it seems some benchmarks are missing (RoseTTAFold, OmegaFold, AlphaFold3). I'm happy for the authors to generally claim that your goal is to work on protein language models that might differ from the nature of, say, AlphaFold3, and that your model sizes are different, such an analysis would still be useful to see by how far off the existing research is.

**Other Comments Or Suggestions:**

N/A

**Other Strengths And Weaknesses:**

Other strengths: Many different design choices are proposed as well as evaluated.

Other weaknesses: In some cases, the design choices feel somewhat ad-hoc (i.e. why did you exactly decide to choose to test this design choice as opposed to another one?). It is quite difficult to follow structure-wise and could benefit from greater clarity in terms of the presentation structure (e.g. a summary table linking the whole design choices). Lastly, it would benefit to have more uncertainty estimates in the tables, especially because some of the values highlighted seem to have small gains.

It would also be beneficial if you could provide a greater discussion about whether and how the design space exploration is novel relative to other work (what other work has explicitly done/explored and what is completely novel), especially within a summary table.

**Questions For Authors:**

Described above.

**Relation To Broader Scientific Literature:**

The paper fits into several areas. First, the paper identifies information loss during structure tokenization and propose ways to deal with this by connecting it to a few different literature areas: (i) vector quantization literature; (ii) latent diffusion model literature; and (iii) the more recent bit-wise discrete modeling literature, connecting it to image synthesis by Han et al. (2024). Second, the geometric structure-aware architecture design uses many ideas from AlphaFold2 and AlphaFold3. Third, the representation alignment connects well with knowledge distillation.

Overall, this paper aims to bridge the gap between sequence-based and structure-based models, and demonstrates how techniques from various fields (e.g. computer vision) can be adapted to protein modeling.

**Theoretical Claims:**

There are no proofs. The formulations of diffusion process, learning objectives for DPLM-2 and residual diffusion models, are clear.

---

> ### Author Rebuttal · Authors · 2025-04-01
>
> Many thanks for your comments that have greatly improved our manuscript! We address your concerns as below. We sincerely thank you once again and welcome any further feedback!
>
> > The bit-level and index-level evaluation is really stark. Is there any broader intuition behind this that you could check experimentally?
>
> Index-level prediction is highly challenging because small changes at the bit level can result in drastically different indices, as shown in the example below. This issue becomes even more problematic as the codebook size increases, further exacerbating the difficulty of direct index prediction.
>
> | Continous struct token | Quantized struct token (bit level) | Index |
> |---|---|---|
> | **+0.1**, -1.5, -3.2, +0.7 | **+1**, -1, -1, +1 | **9** |
> | **-0.1**, -1.5, -3.2, +0.7 | **-1**, -1, -1, +1 | **1** |
>
>
>
> > W1: Some design choices feel somewhat ad-hoc due to the lack of discussions on motivation. It could benefit from greater clarity with a summary table linking the whole design choices.
>
> Thank you for this comment. To make our presentation more structured, we provide a summary [TABLE](https://anonymous.4open.science/r/icml-BF61/Table1.pdf) with greater discussion on all the design choices, traditional methods, motivations, and findings.
>
>
> > W2: It would benefit to have more uncertainty estimates in the tables, especially because some of the values highlighted seem to have small gains.
>
> Thank you for your comment. We are actively launching three more independent runs of the following selected models to provide uncertainty estimates using the mean and the 95% confidence interval (CI). We will include the uncertainty estimates of other variants in the final version.
>
> |  | Cameo2022 |  | PDB Date |  |
> |---|---|---|---|---|
> |  | RMSD↓ | TMScore↑  | RMSD↓ | TMScore↑  |
> | Bit + FM | 6.210 ± 0.100 | 0.840 ± 0.005 | 2.811 ± 0.052 | 0.914 ± 0.010 |
> | Bit + FM + ResiDiff | 6.088 ± 0.085 | 0.844 ± 0.001 | 2.777 ± 0.161 | 0.916 ± 0.005 |
>
> > W3: It would be beneficial to provide a summary table with greater discussion about the novelty of the design space.
>
> Thank you for your suggestion. We provide the summary table above in our response to W1.
>
> > The overall analysis is done on relatively small PLMs (up to 3B). I don't see this as an issue and I understand the constraints of computation resources, but it would be great to acknowledge more explicitly.
>
> Thanks! We acknowledge this statement and will make it explicit in our final version as you suggested. While scaling is effective as shown in DPLM-2, we'd like to reassert that correct design choices are critical. In our work, we show that our 650M multimodal PLM could achieve on par results with the 3B specialized ESMFold on structure folding, showing high parameter efficiency.
>
> > Some missing benchmarks (RoseTTAFold, OmegaFold, and AlphaFold3) on structure prediction tasks could be useful, even though I'm happy for the authors to claim that the focus is on protein language models, which differ from the nature of these benchmarks.
>
> Thanks for your suggestions. We will make a clear claim on our focus on PLMs and will add these benchmarks to the table in the final version.

---

### Official Review · Reviewer_4dFc · 2025-03-20

**Overall Recommendation:** 4

**Summary:**

This paper aims to systematically improve current multimodal protein language models in the following respects: (1) generative modeling, where the authors argue that structural information loss caused by index-based structure tokenization cannot be resolved by improving reconstruction accuracy, and opt for a finer-grained supervision through bit-wise cross entropy loss; (2) geometry-aware architecture and representation alignment to improve higher-order structural modeling, (3) extended data coverage on multimer besides monomer. By doing so, the authors show that the enhanced model built upon DPLM-2 achieves better structural modeling abilities, e.g. reducing the RMSD from 5.52 to 2.36 on PDB test set.

## update after rebuttal
The authors addressed all my concerns. I raised my score to 4.

**Claims And Evidence:**

The claims in this paper are supported by convincing evidence.

**Essential References Not Discussed:**

N/A

**Experimental Designs Or Analyses:**

I have checked the experimental designs, but some settings are unclear to me. Please see Q2.

**Methods And Evaluation Criteria:**

This paper focuses on structure modeling ability of multimodal PLMs, so I think evaluation on structure prediction in RMSD / TMscore suits it fine.

**Other Comments Or Suggestions:**

- L92 "DPLM-2: An multimodal extension of DPLM" -> "A multimodal"
- Table 4 caption is a bit confusing.

**Other Strengths And Weaknesses:**

Strength:
- This paper explores a range of design choices for multimodal protein language models, specifically DPLM-2, which is informative and complementary to the previous study.
- The authors conduct an in-depth preliminary study on structure tokenization and prediction by LM.
- Detailed and insightful ablation studies.

Weakness:
- This paper would benefit from a more structured organization of its contributions to the design space of multimodal protein language models. Rather than presenting incremental improvements in a fragmented manner (as "DPLM-2 + X" throughout different sections), the authors should consider providing a comprehensive taxonomy of the design choices explored, highlighting how these choices are orthogonal and can be composable, and positions these innovations within the broader context of multimodal protein models. Given the authors' ambitious goal of elucidating design spaces for multimodal PLMs, a more systematic synthesis of findings would substantially enhance the paper's impact and utility to the community.

**Questions For Authors:**

1. What is the final version of the recommended design choice? It seems to me that the authors separately propose specific improvements to DPLM-2, and the entire paper seems like an extensive ablation study for it. But if all the designs are modular and complementary to each other, why didn't the authors show the overall performance of combining GeoDPLM + FM + RESDiff + folding SFT + REPA altogether?
2. I did not quite understand the setting in Table 4. As far as I know, DPLM-2 already incorporates a diffusion LM in it, so did "+RESDIFF" add another continuous diffusion to it? And how about "+FM"? L240 states that the authors "finetune such a model with flow matching", does that (DPLM-2 + FM) mean the original diffusion LM is finetuned by FM generative objective (both seq and struct), and "w/ folding SFT" denotes the generative objective of structure only (i.e. denoise structure given sequence)?
3. Following this, the so-called "hybrid structure modeling method" seems to rescue the skewed structure modeling ability of DPLM-2, which adopts the pre-trained sequence model DPLM and finetunes for additional structure tokens. This raises the question that direct fine-tuning approach on DPLM may be suboptimal - why didn't authors train from scratch instead of inheriting DPLM-2 parameters? Wouldn't it yield even better performance if both modalities are jointly considered throughout training?
4. I am curious about how the representation alignment to folding model boosts structure diversity as well as accuracy. Can the authors elaborate on that? Does this have something to do with the potentially unbalanced training set for original DPLM in terms of protein sequence length?

I would consider raising my score if the authors solve these questions.

**Relation To Broader Scientific Literature:**

The work is a meaningful attempt at extending multimodal diffusion protein language model. Previous work DPLM-2 incorporates structural tokens, where discretization brings the benefit of reduced noise and the potential of scaling. This paper builds on top of DPLM-2, and proposes several improvements in advancing its structure modeling ability.

**Theoretical Claims:**

N/A (no theoretical claims in this paper)

---

> ### Author Rebuttal · Authors · 2025-04-01
>
> Many thanks for your comments that have greatly improved our manuscript! We address your concerns as below. We sincerely thank you once again and welcome any further feedback!
>
> > W1: the paper would benefit from a more structured presentation of its contributions (e.g. a taxonomy)
>
> Thank you for your suggestion. We present a comprehensive taxonomy of design choices in a table and discuss their orthogonality and synthesis, as compiled in this [TABLE](https://anonymous.4open.science/r/icml-BF61/Table1.pdf).
>
> > Q1: What is the final recommended design choice? (e.g., combining GeoDPLM + FM + RESDiff + folding SFT + REPA altogether.)
>
> We structured the paper to clearly present the impacts of each design. To provide a complete picture, we now include **new results** that combine all design methods.
>
> |  | PDB Date |  | Cameo2022 |  | Uncond. Gen. |
> |---|---|---|---|---|---|
> |  | RMSD↓ | TM↑ | RMSD↓ | TM↑ | Diversity↑ |
> | DPLM-2 650m | 5.307 | .8306 | 7.703 | .7936  | 0.700 |
> | Bit | 3.221 | .9043 | 6.403  | .8380  | 0.825 |
> | Bit + FM | 2.870 | .9099 | 6.183 | .8418 | 0.525 |
> | Bit + FM + ResDiff (w/o SFT)  | 2.788 | .9146 | 6.077  | .8456 | 0.525 |
> | Bit + FM + SFT + ResDiff | 2.370  | .9270 | 5.847  | .8442  | N/A |
> | New results |  |  |  |  |  |
> | Geo + Bit | 2.551 | .9254 | 5.955 | .8520 | 0.900 |
> | Geo + Bit + FM | 2.443 | .9261 | 6.172 | .8404 | 0.575 |
> | Geo + Bit + REPA | 2.507 | .9264 | 6.192 | .8412 | 0.875 |
> | Geo + Bit + REPA + SFT | 2.404 | .9322 | 5.754 | .8424 | N/A |
> | Geo  + Bit + FM + SFT + REPA + ResDiff | 2.379 | .9297 | 6.200 | .8398 | N/A |
>
> **SFT**: Folding SFT improves the structure folding but sacrifices the model's ability for multimodal co-generation, as it is fine-tuned specifically for the folding task. For models finetuned with the folding SFT objective, we skipped the evaluation of unconditional co-generation.
>
> **The recommended setting** — Geo + Bitwise. Geometric architectures are compatible with bitwise modeling and their combinations achieve comparable results with models finetuned with folding SFT on structure folding, and further obtains an effective improvement on unconditional generation quality & diversity. This setting is also effective in terms of training efficiency as it avoids additional computational overhead from other methods like FM and REPA.
>
> **REPA and bit-based modeling**. Both REPA and bit-based modeling enhance structure folding and generation diversity. Meanwhile, as shown in the table, their combinations do not lead to further improvements. We suggest that this is because REPA and bit-based modeling both help through enabling smooth and high-dimensional learning signals compared to index-based discrete tokens, hence their non-orthogonal effects.
>
> **Hybrid modeling (w/ FM) and geo module**. FM effectively improves folding, but the benefits diminish with Geo modules, and can reduce generation diversity due to its ODE nature. However, benefitting from the same nature, FM accelerates the sampling process by requiring 10x fewer sampling steps.
>
> **ResDiff**. Similar to the results in the paper, ResDiff does not bring a significant boost to folding metrics. The major benefit of ResDiff is to provide a finer local structure as discussed in the Fig. 7 of our Appendix.
>
> > Q2: Clarifications on experimental configurations in Table 4 (+ResDiff, +FM, and w/SFT).
>
> We elaborate on our experiment settings, as compiled in this [TABLE](https://anonymous.4open.science/r/icml-BF61/Table2.pdf)
>
> > Q3: Hybrid structure modeling method seems to rescue the skewed structure modeling ability of DPLM-2, which adopts the pre-trained sequence model DPLM and may be suboptimal compared to training from scratch.
>
> Thanks for raising this insightful discussion. We conducted experiments as you suggested
>
> |  | Cameo2022 |  | PDB Date | |
> |---|---|---|---|---|
> |  | RMSD↓ | TMScore↑ | RMSD↓ | TMScore↑ |
> | Bit & FM from pretrained DPLM-2 | 6.1825 | 0.8414 | 2.8697 | 0.9099 |
> | Bit & FM from scratch | 10.9815 | 0.7090 | 6.4453 | 0.8070 |
>
> As shown, DPLM-2 initialization outperforms training from scratch (see table), likely because: (1) sequence pretraining improves structural modeling (cf. ESMFold); (2) Sequence data vastly outnumbers experimental structures, making pretraining more effective.
>
> > Q4: On REPA boosting diversity and accuracy? Is it related to the unbalanced training set for the original DPLM?
>
> We agree that the unbalanced DPLM training set may contribute to REPA's benefits. REPA further aids structure generation by:
>
> - ​Overcoming discrete token limitations: Quantization loses finer details (Table 1), while representation alignment helps preserve structural nuances via smooth, informative and high-dimensional learning signals.
>
> - ​On learning diffusion model: REPA paper points that high-quality representations are key to diffusion models; we bridge this gap by transferring structural features from folding models.

---

> > ### Comment · Reviewer_4dFc · 2025-04-02
> >
> > I really appreciate the authors' efforts in solving my concerns. I will raise my score to 4 in hopes that this paper gets accepted, and I recommend the authors to include these important clarifications in their revision.

---

> > > ### Author Response · Authors · 2025-04-02
> > >
> > > Dear Reviewer 4dFc,
> > >
> > > Thank you for reading our rebuttal and we sincerely appreciate your supportive feedback and the increased rating! We would like to once again thank you for your insightful comments, and we will surely make more efforts to include these clarifications, discussions and new results in the final version.
> > >
> > > Best,
> > >
> > > Authors

---

### Decision · Program_Chairs · 2025-05-01

**Decision:**

Accept (spotlight poster)

**Comment:**

This paper explores ways to improve multimodal protein language models (PLMs) that combine protein sequences and structures. The authors identify two key challenges: (1) loss of structural information when converting 3D structures into tokens and (2) difficulty in accurately predicting these tokens. To address these, they propose finer-grained supervision using bit-wise cross-entropy loss, better alignment between sequence and structure representations, and the use of data that includes both single proteins (monomers) and complexes (multimers).

All reviewers vote for accept.